# An Experimental Study of Nailed Soil Slope Models: Effects of Building Foundation and Soil Characteristics

Mahmoud H. Mohamed, Mohd Ahmed * and Javed Mallick 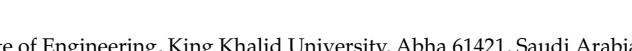

Civil Engineering Department, College of Engineering, King Khalid University, Abha 61421, Saudi Arabia; m_hussein71@yahoo.com (M.H.M.); jmallick@kku.edu.sa (J.M.)
* Correspondence: mall@kku.edu.sa; Tel.: +966-172-418-439

**Abstract:** A soil nailing system is a proven effective and economic method used to stabilize earth slopes from the external (factors increasing the shear stress) and internal (factors decreasing material strength) failure causes. The laboratory models with scales of 1:10 are used to study the behavior of nailed soil slope with different soil and building foundation parameters. The models consist of Perspex strips as facing and steel bars as a nailing system to increase the stability of the soil slope. The models of sand beds are formed using an automatic sand raining system. Devices and instruments are installed to monitor the behavior of soil-nailed slope during and after construction. The effect of the soil type, soil slope angle, foundation width and position on the force mobilized in the nail, lateral displacement of the slope, settlement of the foundation and the earth pressure at the slope face, under and behind the soil mass at various foundation pressures, has been observed. It is found that the increase of soil density reduces both slopes facing displacement and building foundation settlements. The slope face displacement and footing settlement will increase with an increase in the width of the foundation and foundation position near the crest of the slope.

**Keywords:** soil nailing system; soil characteristics; soil slope; foundation pressure; nailed soil

## 1. Introduction

The failure of slopes leads to extensive social and economic consequences and is accompanied by the degradation of the natural environment. A soil nailing system is a proven effective and economic method used to stabilize earth slopes from the external (factors increasing the shear stress) and internal (factors decreasing material strength) failure causes. A review of the literature related to the susceptibility of slope failure is presented by Pourghasemi et al. [1]. The comprehensive review of ground improvement techniques using earth nailing systems, including system installation and construction, failure modes of soil-nailed structures and effects of various construction parameters on the nailed soil slope design method, is presented by Sharma et al. [2]. The guidelines are suggested by Yelti [3] to improve the design procedures for soil-nailed structures considering the lateral pressures generated by soil expansion. The soil nailing system is used to stabilize earth slopes by installing closely spaced nails in the ground in situ. A soil-nailed slope essentially consists of three elements, namely, the soil mass, the slope facing and steel reinforcement (nails). The deformations behavior of the soil nailing system depends on the composite interactions between the soil nailing system elements. The slope facing provides the local stability of the ground between the nails and limits its decompression in the soil nailing system [4]. Bridges et al. [5] have proposed a measurement procedure for the crest displacement of the soil-nailed wall and concluded that effective soil cohesion has the greatest effect on crest displacement. Kotake and Sato [6] have studied the influence of bending stiffness of facing material, installed on a nailed slope surface of dense sand, on deformation and bearing capacity characteristics by conducting the tests on rigid and flexible models of footings. Various experimental and numerical models of nailed soil structures have been studied for the evaluation of structural performance and

the development of reliable design methods for nailed soil structures. Ceccato et al. [7] have developed numerical models based on the material point method to investigate the application of plate anchors for landslide stabilization. The numerical model has been validated with the results of some small-scale laboratory tests. Sahoo et al. [8] have carried out the shaking table tests to study the seismic behavior of nailed soil slopes. A simplified limit equilibrium method is presented by Deng et al. [9] to analyze the stability of slopes reinforced with anchor cables considering nonlinear Mohr–Coulomb strength criterion for the shear failure of the slip surface. The laboratory models have been developed by Mahmoud et al. [10] to study the effects of surcharge loading and nails characteristics in Nailed Soil Slope behavior. They found that the increase in length and inclination of soil nails decreases the vertical, horizontal stress and footing settlement, while the increase of spacing of nails increases the vertical and horizontal stress behind the soil mass.

Babu and Singh [11] have studied the lateral displacements, nail forces and failure modes of nailed soil vertical slope under both static and seismic conditions. Numerical analysis of stabilizing mechanisms of loose-fill slopes using hybrid nail arrangement has been carried out by Cheuk et al. [12], who studied the influence of hybrid nail orientations on the behavior of the ground nailing system and concluded that a hybrid nail arrangement would enhance the robustness of the ground nailing system. Taule [13] has applied Bishop's Simplified method to evaluate and rank the sensitivity of the soil cohesion, unit weight and internal friction angle with respect to the global factor of safety using Monte Carlo simulation in a soil nailing design. He has inferred that the soil internal friction angle is the most sensitive parameter, while cohesion and unit weight are less sensitive with respect to the global safety factor. Villalobos et al. [14] present the re-assessment stability analysis of soil nailing design and construction in a heavily weathered granite (residual soil) using a limit equilibrium sliding block method (bi-linear failure surface). They found that even after a maximum acceleration of 0.63 g of a stronger earthquake, the nailed wall did not show any damage, probably due to the use of un-drained shear strength parameters. Alhabshi [15] has studied the hybrid mechanically stabilized earth and soil nail structure and has proposed the optimum length of nails required to reduce the structure displacement. A slope stability analysis of a combined system of soil nails and stabilization piles on loess soil is presented by Wu et al. [16]. Benayoun et al. [17] have proposed the genetic algorithm-based optimization of the soil-nailed structure design. Chen et al. [18] have developed the novel element nail pull-out test to describe the shearing mechanisms at the soil–nail interface in nailed cohesive soils. The stability and displacement characteristics of a composite silt–soil-nailed symmetrical foundation pit are numerically investigated by Han et al. [19]. They have found the critical values of inclination and spacing of the soil nails, the diameter and embedded depth of the mixing pile for the stability of the foundation pit. The soil slope deformation pattern is investigated by Sojoudi and Sharafi [20] using the numerical and experimental modeling of layered and homogenous piled stabilized earth slopes. They study the impact of pile position and surcharge distance from the slope crest on soil deformation patterns, the bearing capacity improvement ratio and slope stability improvement ratio.

Various parameters affect the behavior of the earth nailing systems, such as nail parameters, method of installation of nails, soil density, soil slope height and angle, foundation parameters and loadings, among others. It is clear from the literature survey that the effect of the building foundation parameter on nailed soil slope has not been given due attention. The present paper has investigated the performance of soil-nailed slopes laboratory models under varying soil and building foundation parameters. The effect of the type of soil, soil slope angle, footing width and position of footing on the force mobilized in the nail, lateral displacement of the slope, settlement of the footing and the earth pressure at the slope face, under and behind the soil mass at various foundation pressure have been observed. The results for lateral displacement and footing settlement are presented in terms of the percentage of slope height.

## 2. Laboratory Models

The three-dimensional small-scale laboratory models of a dry sand slope with a steel nailing system were developed to predict the performance of soil nailing system behavior. The physical models, their dimensions, state of stress and strain and enclosing walls are built and designed so that their performance perfectly simulates the prototype behavior. The scale factor of 10 is adopted in simulating the soil slope with consideration to material handling. The physical models under strip loading are assumed to develop plane strain conditions. The developed physical models have no influence of soil enclosure boundaries on the behavior of the soil nailing system elements, i.e., soil, nails, facing wall and the footing. The depth and width of the experimental model of nailed soil slope also fulfill the criteria that pressure isobars of strip loading laterally extend almost two times the footing width from the center line of the footing and vertically extend five times the footing width.

### 2.1. Simulation and Dimensional Analysis

The stiffness parameter for the facing wall and nails are maintained for both model and prototype to achieve the correct similarity as per the following relationship.

$$(H^4/EI)_{model} = (H^4/EI)_{prototype} \tag{1}$$

where: $H$ = Slope height (m); $E$ = Young's modulus of elasticity of wall or nail (kPa); $I$ = Moment of inertia per unit length of the wall or the nail (m$^4$/m).

If we consider a scale factor ($N$) = ($H_{prototype}/H_{model}$) and in the case of using the same material for the model and prototype, the relation between the model and prototype becomes:

$$I_{model} = I_{prototype} (E_{prototype}/E_{model})(1/N)^4 \tag{2}$$

Considering the same unit weight and shearing resistance in the model and prototype soil, then the length and spacing between the nails, the distance of footing from the crest of the slope and applied footing pressure of the model as well as the resulting settlements are proportional to the prototype with the same ratio as the total excavation height ratio.

### 2.2. Materials, Testing Tank and Loading Frame

The main components of the nailed soil slope system are back-filling material, nails, facing unit and footings. The air-dried siliceous sand is used in the construction of the soil slope model. The model in layers of sand is formed using an automatic sand raining system to control the density. The properties of the sand were determined as per ASTM standards. The procedure for determining the properties of sand is discussed in the literature [21]. Three sand types are used in the model, namely, loose sand with a relative density of 35% and internal angle friction of 30°, medium sand with a relative density of 48% and internal angle friction of 34°, and dense sand with a relative density of 69% and angle internal friction of 40°. The physical properties of the soil material (sand) are given in Table 1. The facing unit in the nailed soil system only has a minor mechanical role, and the main function of the facing is to ensure local stability of the soil between the nails and to limit its decompression. The facing materials should be flexible enough to withstand ground displacement during excavation. Therefore, a Perspex plate with a thickness of 5.0 mm is selected to simulate the Shotcrete in the prototype soil nailing system of thickness 140.0 mm. The nails used in this study are circular cross-section steel bars with a length of 700 mm. The diameter of the model nails is 5 mm to simulate the actual nail diameter of 50 mm. A direct tension test is conducted to know the mechanical properties of the nail [21].

**Table 1.** Physical properties of soil.

| Property | Value | Property | Value |
|---|---|---|---|
| % of clay | 0.00 | Coefficient of uniformity ($C_u$) | 1.99 |
| % of silt | 1.33 | Coefficient of gradation ($C_u$) | 1.00 |
| % of fine sand | 39.17 | Specific gravity ($G_s$) | 2.62 |
| % of medium sand | 58.63 | Minimum unit weight ($\gamma_{min}$) (kN/m$^3$) | 15.30 |
| % of coarse sand | 0.87 | Maximum unit weight ($\gamma_{max}$) (kN/m$^3$) | 17.80 |
| % of fine gravel | 0.00 | Minimum void ratio ($e_{min}$) | 0.472 |
| Effective diameter ($D_{10}$) mm | 0.126 | Maximum void ratio ($e_{max}$) | 0.712 |

Considering the simulations criteria for the material and dimensions of the tank, an open-front Perspex box measuring 1760 mm × 850 mm × 1000 mm is chosen as the experimental model of nailed soil slope. The specially designed loading system is used in this work to prevent any disturbance to the nailed soil model and is shown in Figure 1. The loading system is made of a loading frame, loading frame base and hydraulic loading system. A rigid steel plate of dimensions 840 mm × 150 mm × 22 mm thick is used to act as a building foundation to exert a distributed load on the soil. The nailed soil model in the testing box has used a total of 2250 kg of dry sand and nine nails. The mechanical properties of nails, facing and building foundation are given in Tables 2 and 3.

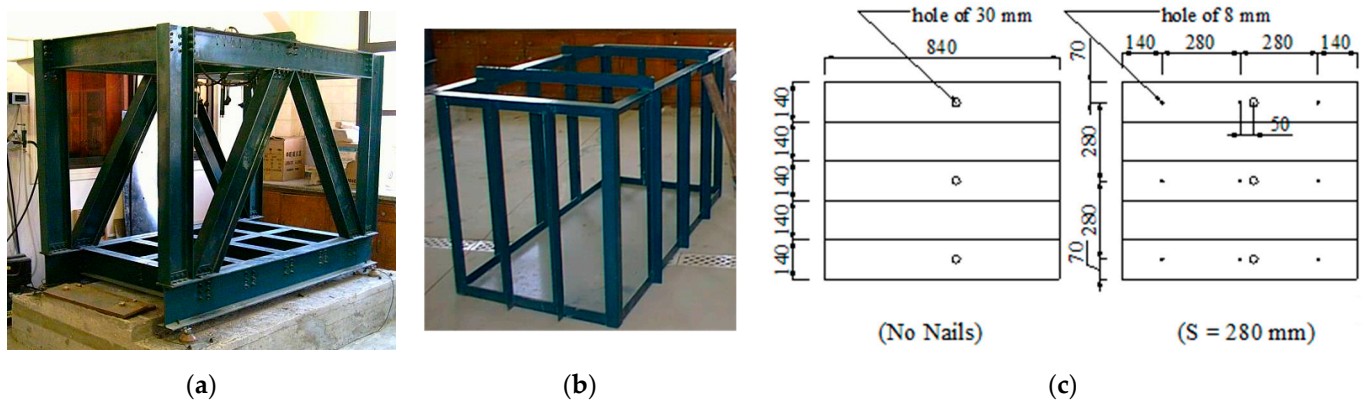

|        (a)        |        (b)        |        (c)        |

**Figure 1.** Testing loading frame, sand tank and slope facing panels; (**a**) loading frame; (**b**) sand tank; (**c**) slope facing panels.

**Table 2.** Nail properties.

| Element | Length (mm) | Maximum Tensile Force (kN) | Strains at Ultimate Stress (με) | Maximum Tensile Strength (kPa) | Young's Modulus (kPa) | Flexural Rigidity, EI (kN.mm$^2$) | Normal Stiffness, EA (kN/m) |
|---|---|---|---|---|---|---|---|
| 5 mm Steel Nails | 700.0 | 17.4 | 7625 | $8.8 \times 10^5$ | $21.2 \times 10^7$ | $6.51 \times 10^{-3}$ | 4165.9 |

**Table 3.** Slope facing and foundation plate properties.

| Plate Element | Material | Width (mm) | Thickness (mm) | Young's Modulus (kPa) | Bending Stiffness (kN.mm$^2$) | Axial Stiffness (kN) |
|---|---|---|---|---|---|---|
| Footing | Steel | 150 | 22.0 | $21.2 \times 10^7$ | 188,186.6 | 4665.78 |
| Facing | Perspex Plate | 140 | 5.0 | 4200 | 0.04375 | 0.021 |

*2.3. Experimental Program*

2.3.1. Preparation of Sand Filled Tank with Reinforcement

1.  The 150 mm thick first layer of the sand is formed using the sand spreader and leveled as desired to obtain the required predetermined density.
2.  The bottom layer nails with tubes are fixed in their positions.
3.  Then, the second layer is formed, and the layer is compacted to reach the required density.
4.  In each layer, the horizontal and vertical pressure cells were installed in their predetermined positions.
5.  Steps No. 2, 3, 4, were repeated until the tank is filled up and ready to begin the test.

2.3.2. Preparation of Sequence of Construction Phases

The experimental set-up is planned to match the sequence of slope construction in field, i.e., excavation of soil, construction of slope soil nailing system and subsequent placement of foundation pressure. To simulate the actual behavior of the prototype model, the test was divided into two stages. The first stage was the excavation stage to make sand slope, and the second was the loading stage.

1.  The 140 mm thick layer of sand was excavated.
2.  The first facing panel is placed that includes pressure cells in line with the middle of the panel, and this facing panel is held with two polyethylene strips (sticky tape) to the box side to prevent sand from dropping from the gap between the edges of the slope facing and the box sides.
3.  Then, the installation of the first row of strain gauge fitted nails is carried out, which is tightened to the slope facing with nuts by pushing the nail into the sand. These nails are placed in their position using the tubes (black lines in Figure 2).
4.  Dial gauges are fitted to record the displacements of the slope facing due to excavation of the second layer.
5.  Another four layers of the sand, each 140 mm thick, are dug, and the facing panels are placed as before. At the third and fifth layers, a row of nails and pressure cells are fitted. The facing panel details are shown in Figure 1.
6.  In each stage, displacement of the slope face, tensile force in the nails and pressure (horizontal and vertical) are recorded.
7.  As the excavations are complete to a depth of 700 mm, the rigid footing was placed at the required location from the nailed slope crest at the top surface, leveled by a water bulb level and then loaded to the required pressure. The different stages of nailed soil slope constructed are depicted in Figure 2.
8.  The settlement of the footing is measured using LVDTs.
9.  The strain gauge reading is recorded simultaneously with the LVDTs and pressure cells for each foundation pressure case. The spreading of pressure in the nailed soil, horizontal stresses on the slope facing and the horizontal pressure behind the nailed soil are measured by means of five soil pressure transducers and six miniature strain gauge cells, namely, local cell pressure transducers.

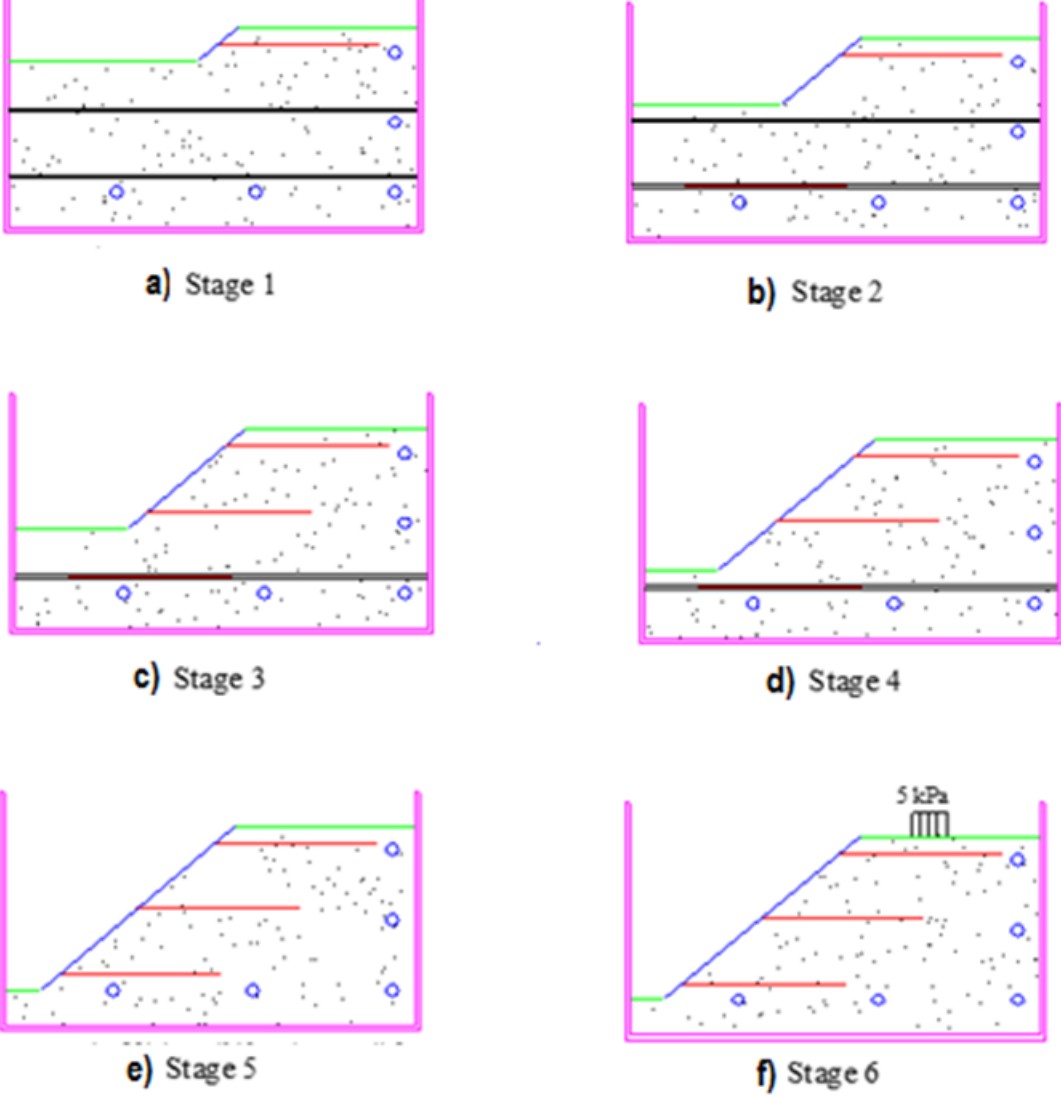

**Figure 2.** Slope-nailed system construction stages [21].

## 3. Results and Discussion

In the present study, the laboratory models are prepared with three soil types with different relative densities, i.e., 35%, 48%, 69%, with two angles of slope ($\theta$), i.e., 40°, 45°, with two footing widths (B), i.e., 150 mm and 200 mm and with four positions of footing from upper slope corner (X), i.e., 75 mm (0.5 B), 150 mm (B), 225 mm (1.5 B), 300 mm (2 B). The displacement of the slope, force in the nail, settlement of the footing and the earth pressure in back-fill soil due to different soil and footing parameters (relative density, slope angle, footing width, footing position) are measured and obtained results are discussed below. The horizontal nails ($\delta = 0$) with a length of slope height ($L_n = 700$ mm) are used in the models. The vertical ($S_v$) and horizontal ($S_h$) spacing of nails are provided as 0.4 times the slope height (0.4 *H*). The lateral displacement of the nailed slope facing are recorded with three dial gauges at three measurement points A, B and C at distances Z = 70 mm (0.1 *H*), 350 mm (0.5 *H*) and 630 mm (0.9 *H*) from the top surface. Pressure cell numbers are mounted flush with the back of the slope facing with a special arrangement to measure horizontal pressure, as shown in Figure 3 at points A, B and C. Readings are measured during construction of slope and after each increment of foundation pressure (q) from 5.0, 10.0, 20.0, to 30.0 kPa. The complete nailed soil models with various parameters are shown in Figure 3.

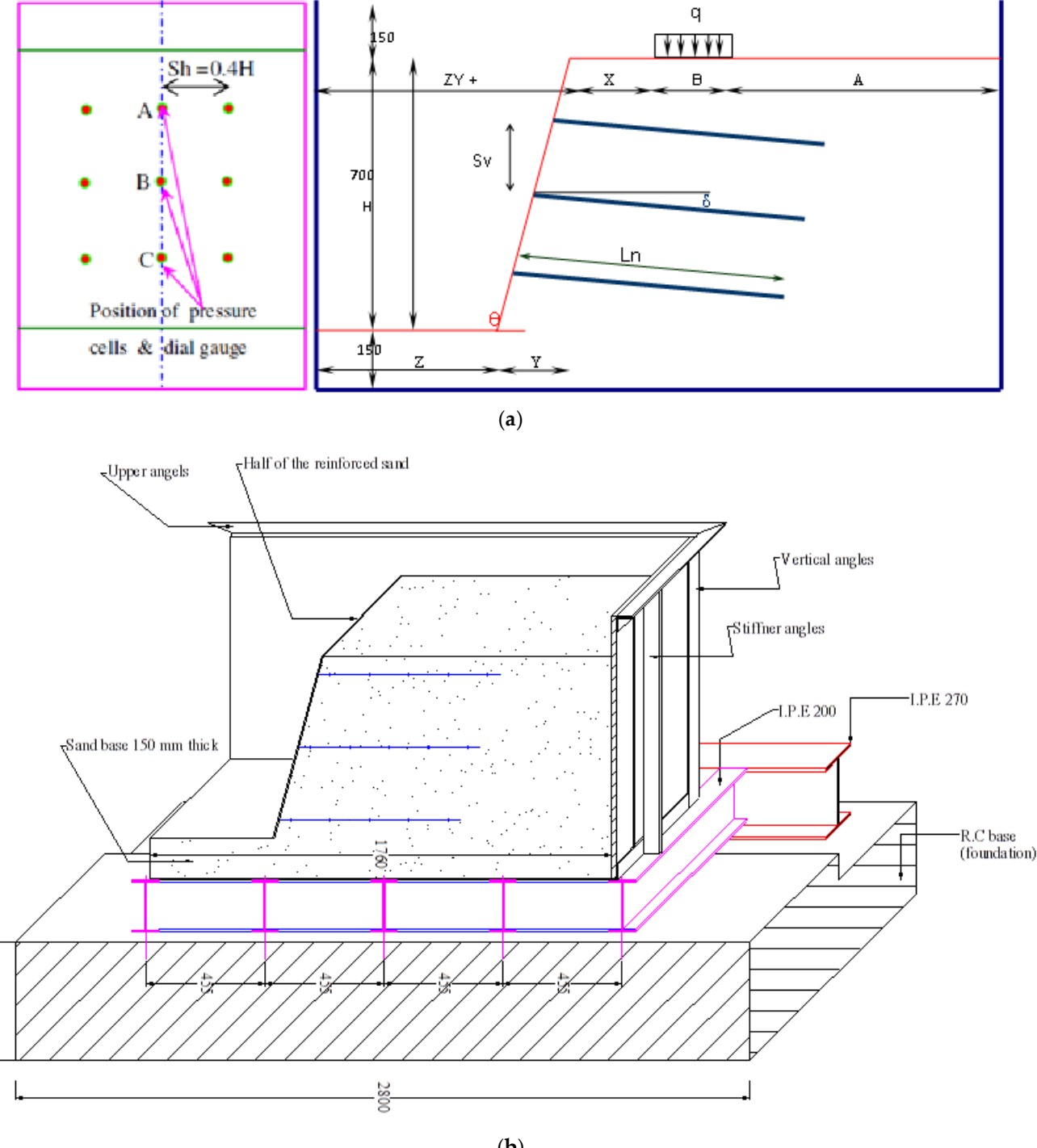

**Figure 3.** Laboratory model with different soil and building foundation parameters; (**a**) section and elevation of nailed soil slope model; (**b**) 3D model of nailed soil slope (all dimensions are in mm).

### 3.1. Effect of Soil Type

3.1.1. Horizontal Movement of the Soil Nailing Slope

The effect of the soil type on the horizontal movement of the slope face of the nailed soil system during the construction stage and loading stages is shown in Figure 4a. It is clear from the figure that the inclusion of reinforcement reduces the horizontal movement of the slope face. The figure also depicts that the horizontal movement of the slope in the middle third is higher than the slope top and bottom levels, and it has the lowest value at

the top of the slope. This is attributed to the boundary condition of the soil slope. The top level of the slope with foundations and bottom portion will restrain the movement of the slope face.

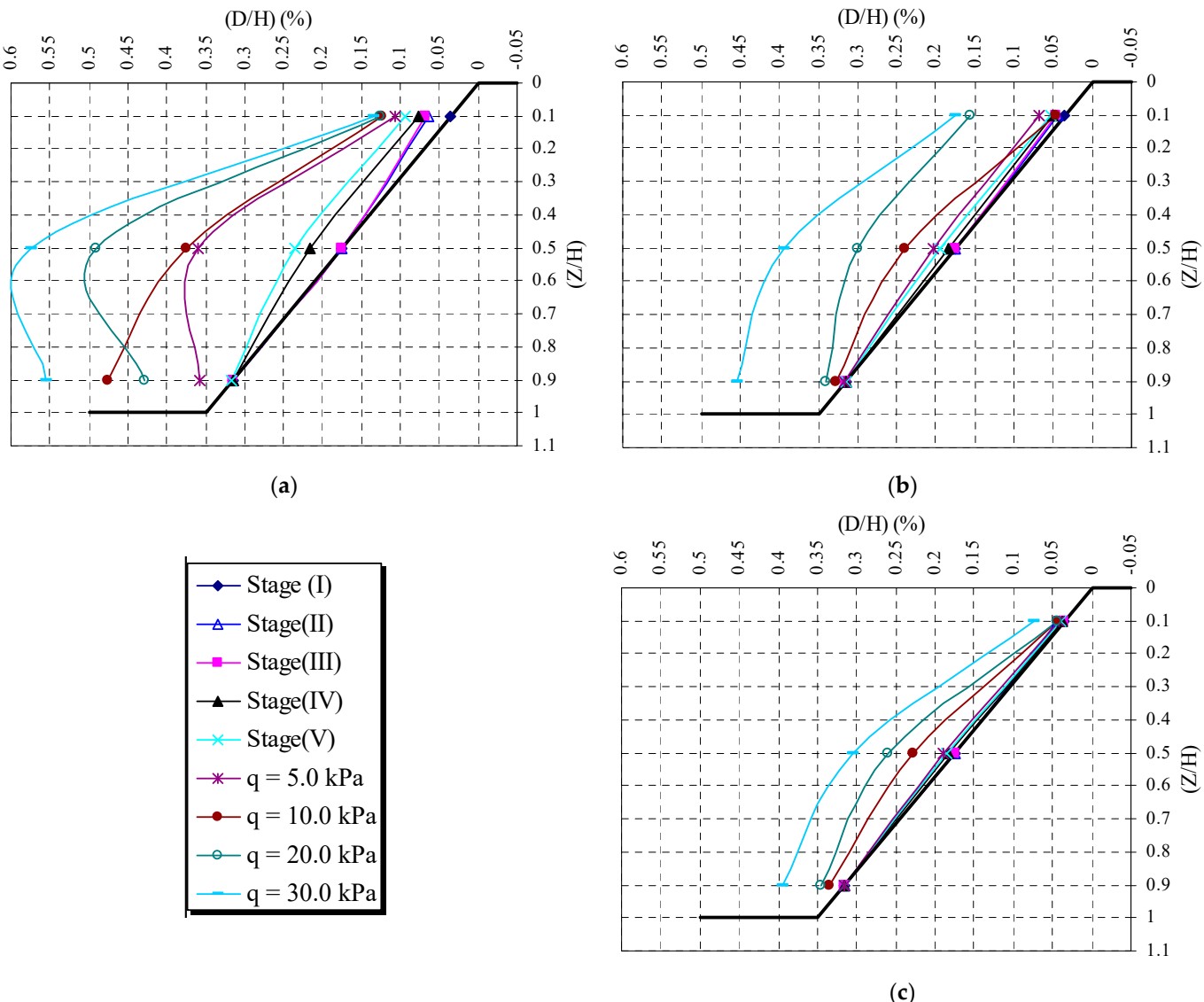

**Figure 4.** Influence of soil density on the horizontal movement of slope face in construction and loading stages; (**a**) loose sand; (**b**) medium sand; (**c**) medium sand.

Figure 5 shows the effect of the soil type on the horizontal movement of the slope face during the construction stage and loading stages of varying foundation pressure. It can be inferred from the results that the increase of the soil density reduces the lateral movement of the soil-nailed slope. Reduction in the slope movement may be because the friction between soil and reinforcement is strongly dependent on the density of the soil and restrained dilatants behavior in dense granular soil. The lateral movement of the slope face of the soil with 69% relative density is less than half of the lateral movement of the slope face of the soil with 35% relative density at higher foundation pressure.

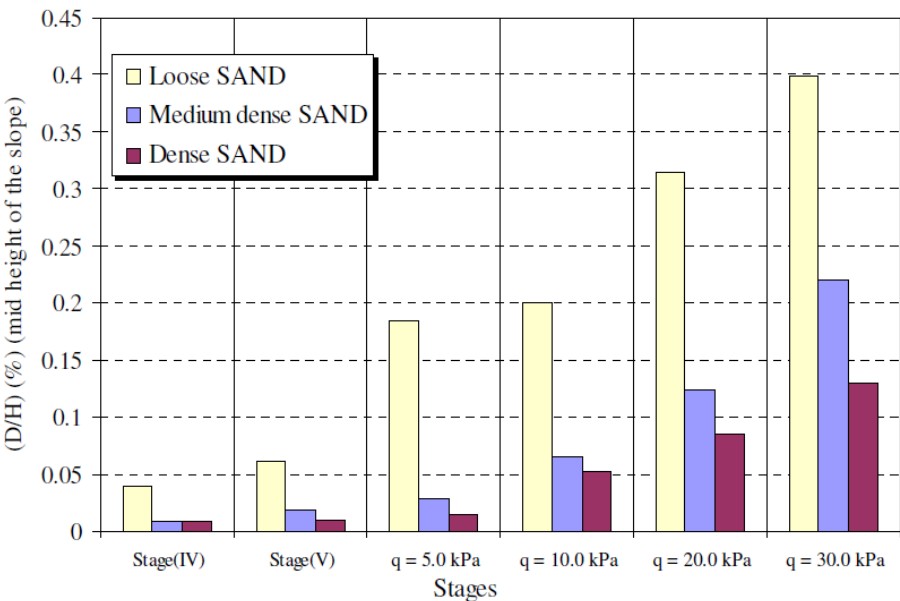

**Figure 5.** Influence of soil density on the horizontal movement at mid-height of slope.

### 3.1.2. Settlement of the Footing

The footing settlements for different soil densities at different footing pressures are presented in Figure 6. It is clear from the figure that the increase of the soil density reduces the footing settlement. This reduction of the settlement could be attributed to the reduction of the facing displacement. Furthermore, the increase of the soil density is accompanied by an increase in shearing resistance and thus a reduction of the footing settlement. The difference of footing settlement of loose sand and dense sand is more than twice at all stages of loading, but the difference of settlement of footing on medium dense sand nailed slope reduces at higher foundation pressure.

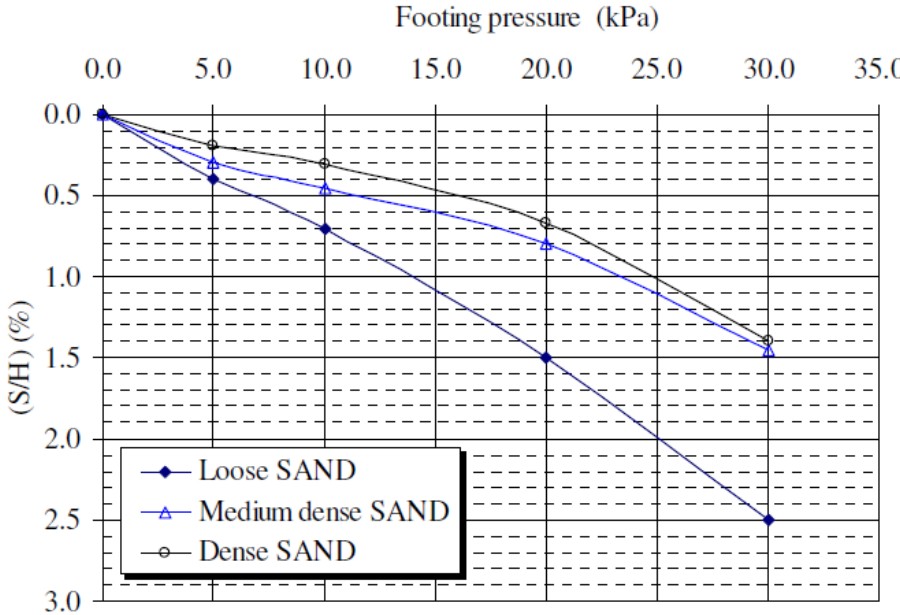

**Figure 6.** Influence of soil density on the settlement of the footing.

### 3.1.3. Force in the Nail

The effect of soil density on the maximum tensile force in the nails at different levels is shown in Figure 7. It is clear from the figure that as the density of the soil increases, the

tensile force in nails both in construction stages and loading stages. This may be because the dense sand is more stable and mobilizes a smaller force than the force in a loose state. It is also clear that the lower nails have less tensile force than the upper nails, but the middle nails have the maximum tensile force. This may be due to the occurrence of maximum horizontal displacements at the mid-height of the slope.

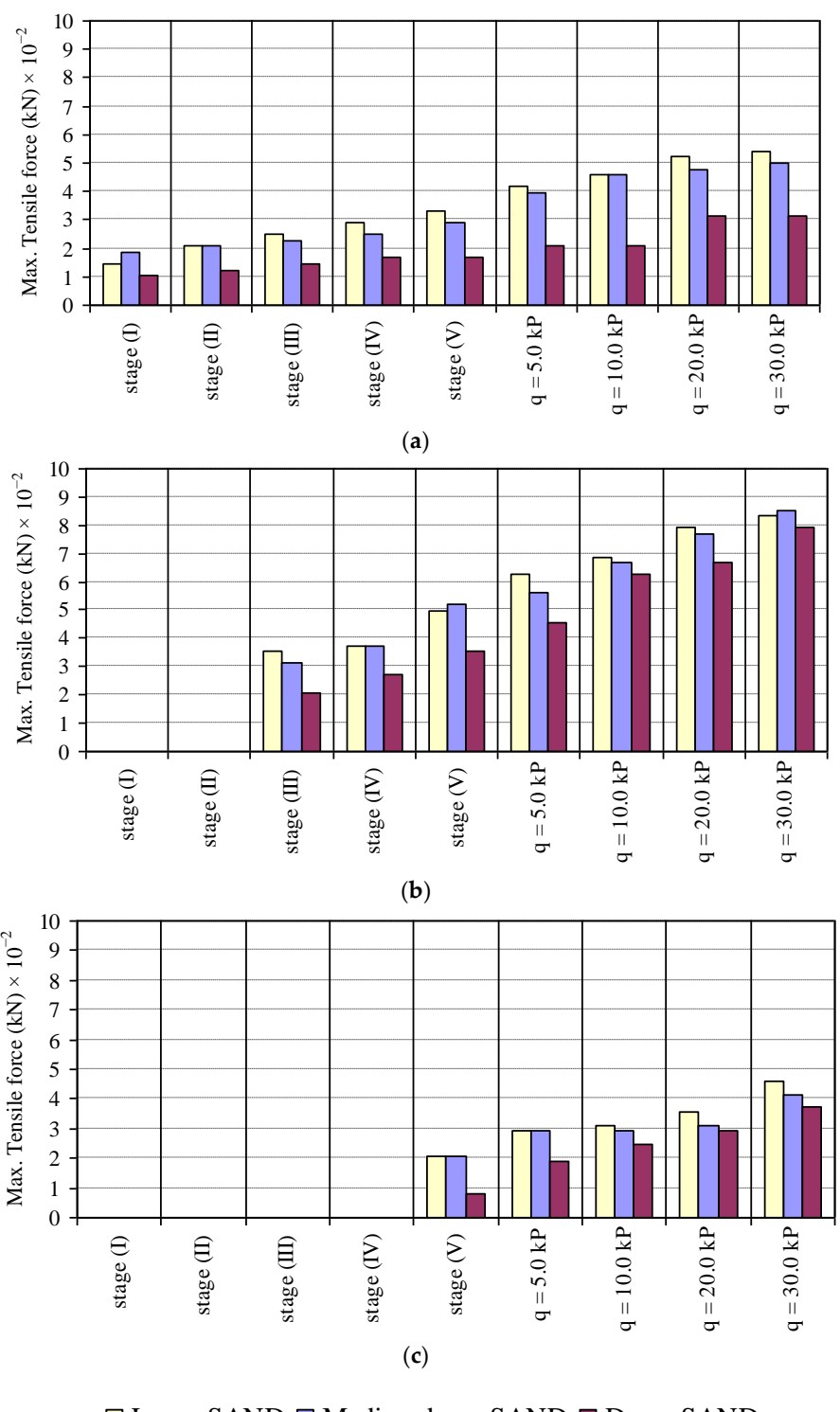

**Figure 7.** Influence of soil density on the maximum tensile force in the nails; (**a**) upper nail; (**b**) middle nail; (**c**) lower nail.

### 3.1.4. Vertical Pressure under Nailed Soil Mass

Figure 8 presents the effect of soil density on vertical stresses under the soil mass at various loading stages. It is evident from the figure that the effect of soil density on vertical stresses under the soil mass was directly proportional, i.e., the denser the soil, the higher the vertical stresses under the soil mass. This may be due to an increase in the unit weight.

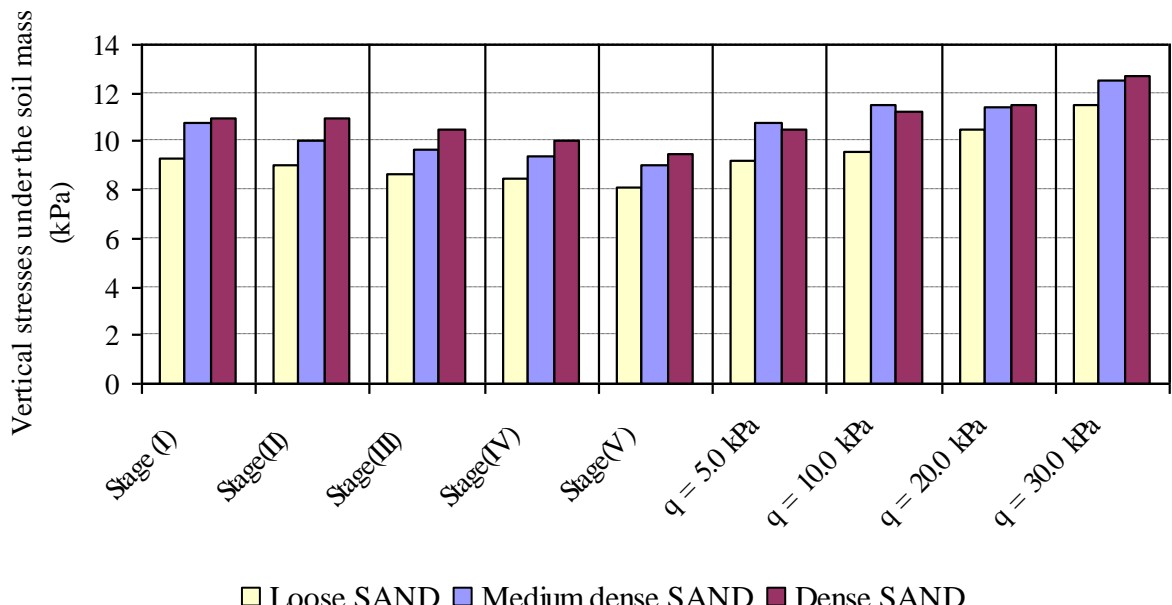

**Figure 8.** Influence of soil density on the vertical stresses under the nails.

### 3.1.5. Horizontal Stresses on the Slope Face

Figure 9 shows the influence of the soil density on the horizontal stresses at the slope face at the middle of the slope during construction stages and loading stages. The figure depicts that, in general, when the soil density increases, the horizontal stresses increase. The percentage increase in horizontal stresses at the slope face increases with the increase of surcharge loading. The percentage increase in horizontal stresses at the slope face subjected to higher surcharge loading is about 20% when the soil density increases from 35% to 69%.

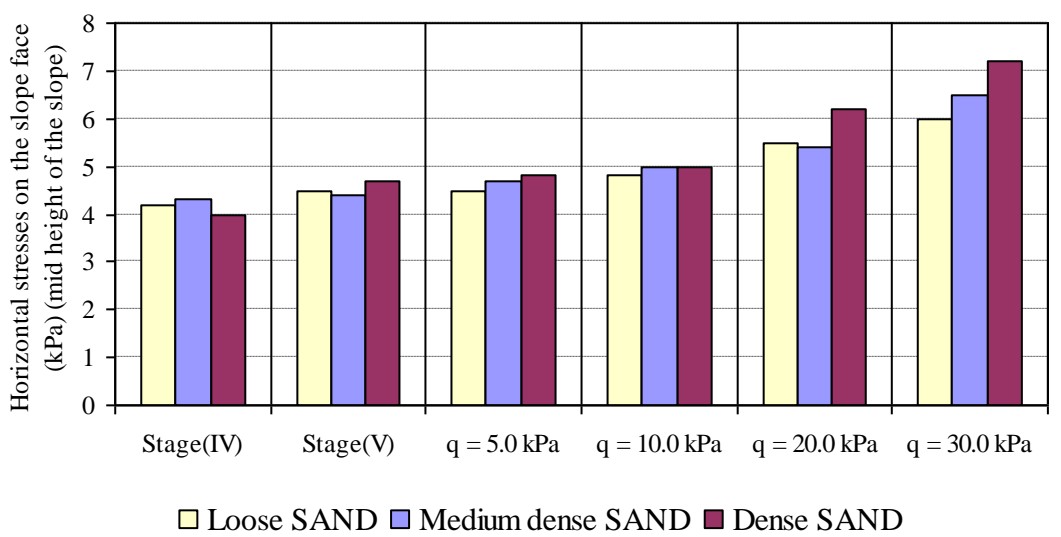

**Figure 9.** Influence of soil density on the slope face horizontal stresses at mid-height.

### 3.1.6. Horizontal and Vertical Stresses behind Reinforced Soil Mass

The horizontal and vertical stress behind the soil mass at different soil densities during construction and loading stages are given in Figures 10 and 11. It is clear from the figures that at construction stages, the vertical stresses behind the soil mass increase as the soil density increases. The behavior of horizontal stresses behind the soil mass with the soil density shows an inverse behavior, i.e., the horizontal stresses behind the soil mass decrease when the soil density increases. A similar trend of variation of horizontal and vertical stresses is also obtained at loading stages. The increase or decrease in horizontal stresses at various stages is more as compared to vertical stresses behind the soil mass at different soil densities.

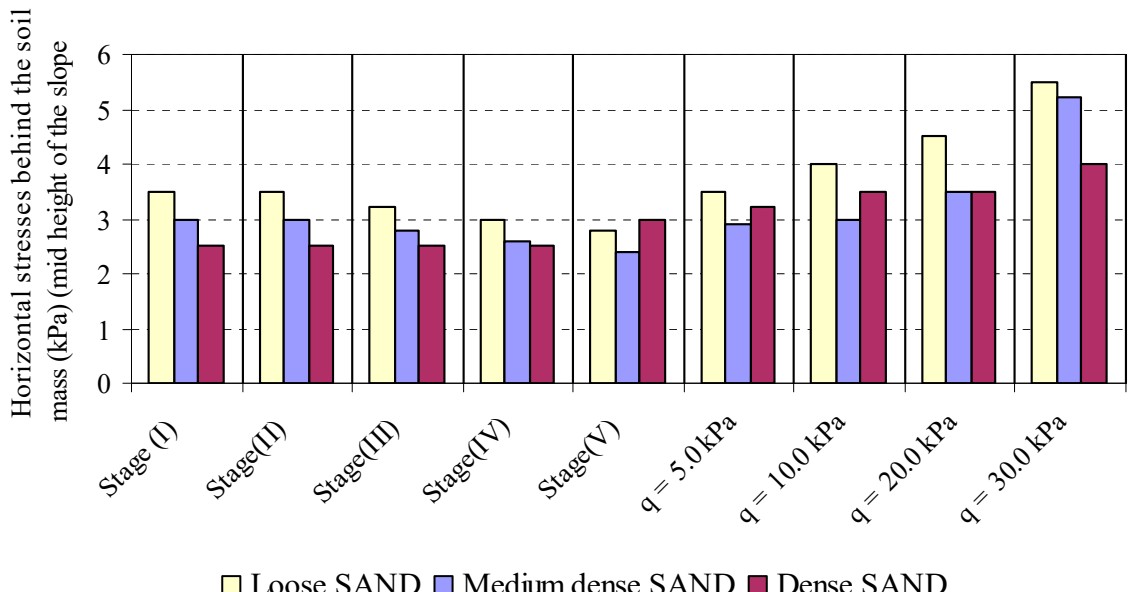

**Figure 10.** Influence of soil density on horizontal stresses behind the nailed soil mass at mid-height of slope.

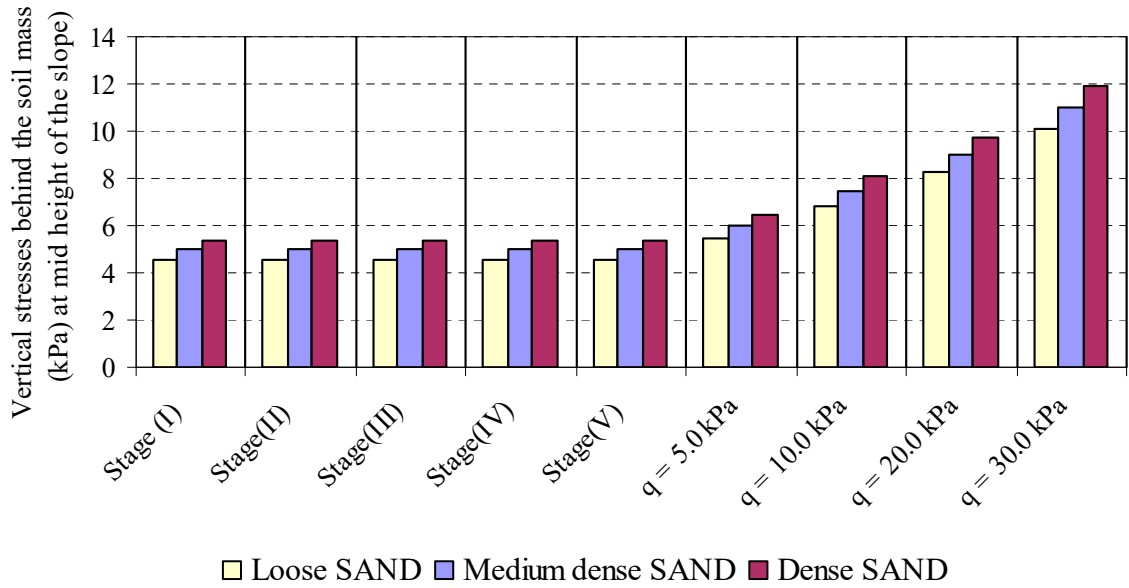

**Figure 11.** Influence of soil density on vertical stresses behind the nailed soil mass.

*3.2. Inclination of the Soil Slope*

3.2.1. Horizontal Movement of the Nailing Slope

Figure 12 represents the effect of soil inclination on the horizontal movement of the slope face during construction stages and loading stages. It is clear from the figures that there is no significant effect of slope inclination on the horizontal movement of the slope face except at heavy surcharge loading.

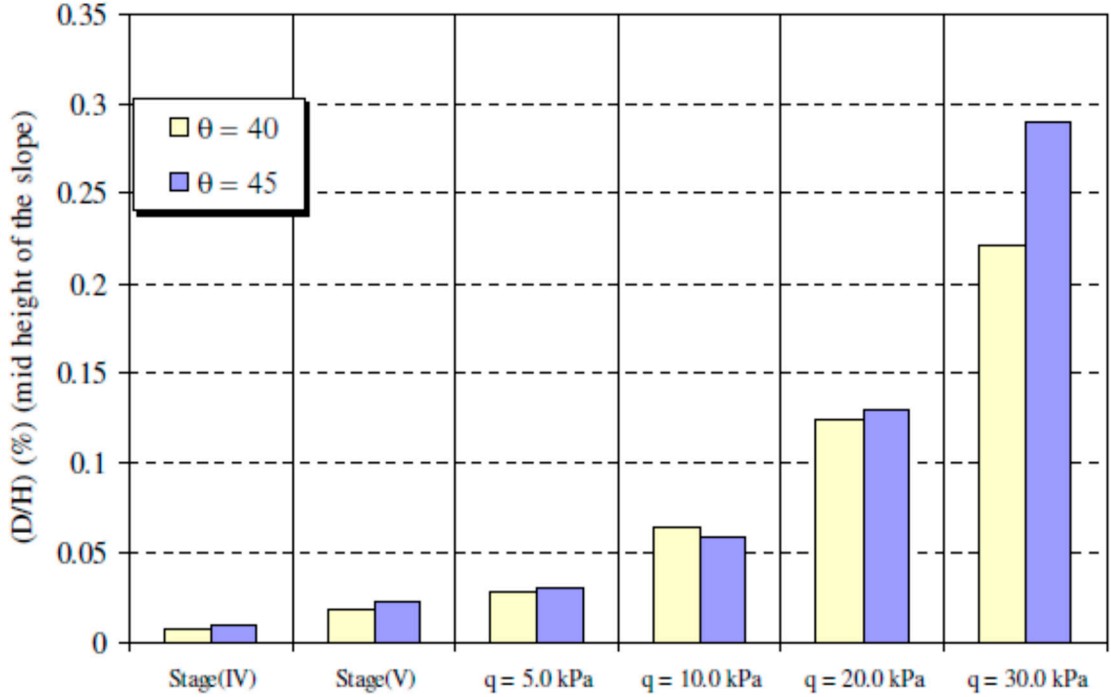

**Figure 12.** Influence of slope inclination on the horizontal movement at mid-height of slope.

3.2.2. Settlement of the Footing

Figure 13 shows the footing settlements for slope inclinations at various footing pressures. It is observed from the figure that the footing settlements have not reduced considerably with the increase of the slope inclination at various surcharge pressure. The reduction of footing settlements with an increase of the slope inclination by a ratio ranged from 20.0% to 37.9%.

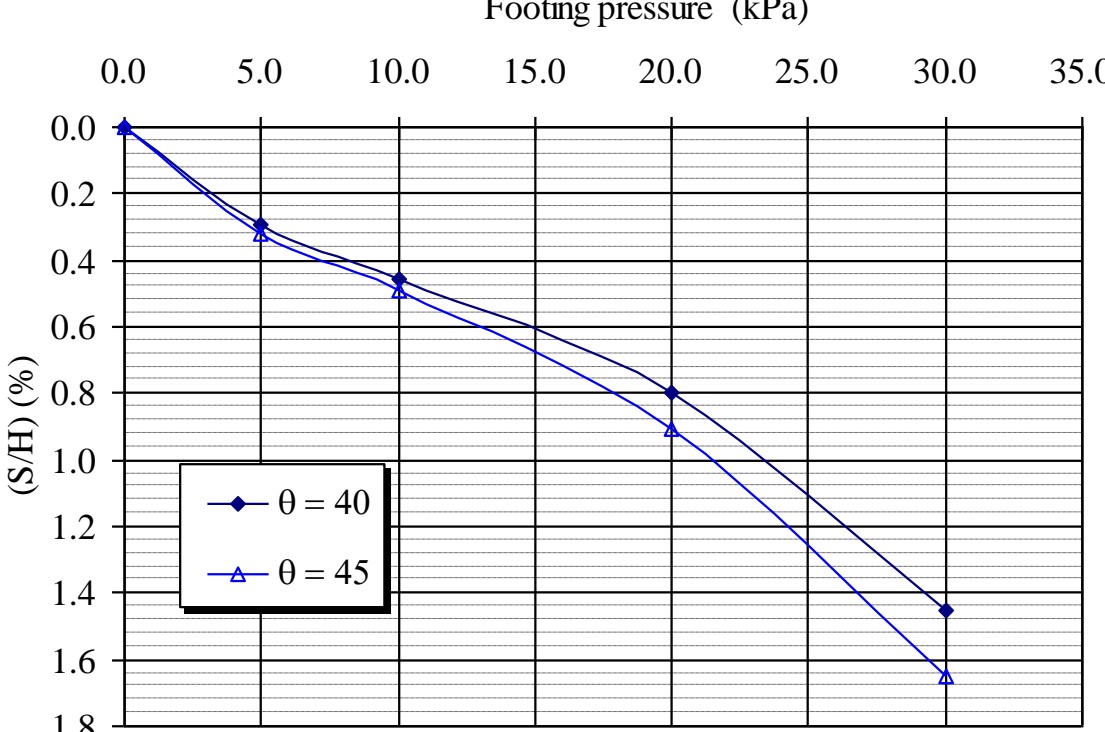

**Figure 13.** Influence of slope inclination on the settlement of the footing.

3.2.3. Force in the Nail

The influence of the slope inclination on the maximum tensile force in the nails at different levels is shown in Figure 14. It is evident from the variation of tensile force that as the slope inclination increases, the maximum tensile force of middle nails decreases in construction stages but increases in tensile force loading stages. The variation of tensile force in the upper and lower nails is different, and the maximum tensile force increases with the increase of slope inclination at all stages. It is also clear that the lower nails have lesser tensile force than the upper nails, and the middle nails have the maximum tensile force. This may be attributed to the maximum horizontal displacements at mid-height of the slope.

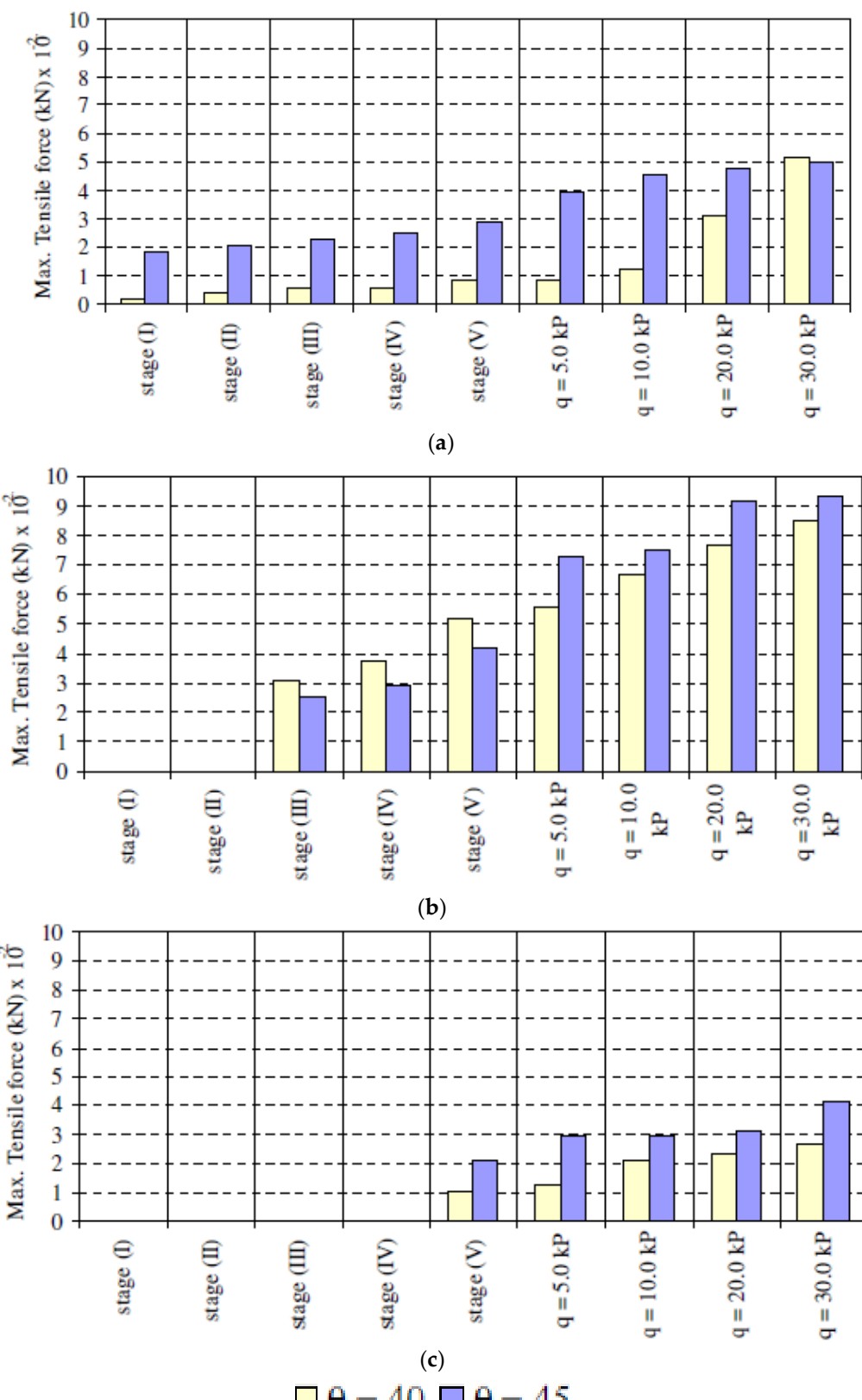

**Figure 14.** Influence of slope inclination on the maximum tensile force of the footing; (**a**) upper nail; (**b**) middle nail; (**c**) lower nail.

### 3.2.4. Horizontal Stress on the Slope Face

Figure 15 represents the influence of the slope inclination on the horizontal stresses at the slope face at the middle point of the slope during construction stages and after applying footing pressure. It is clear from the figure that as the slope inclination increases, the horizontal stress at the slope face increase in both construction and loading stages. The percentage increases in the horizontal stresses at the slope face range from 16% to 43%.

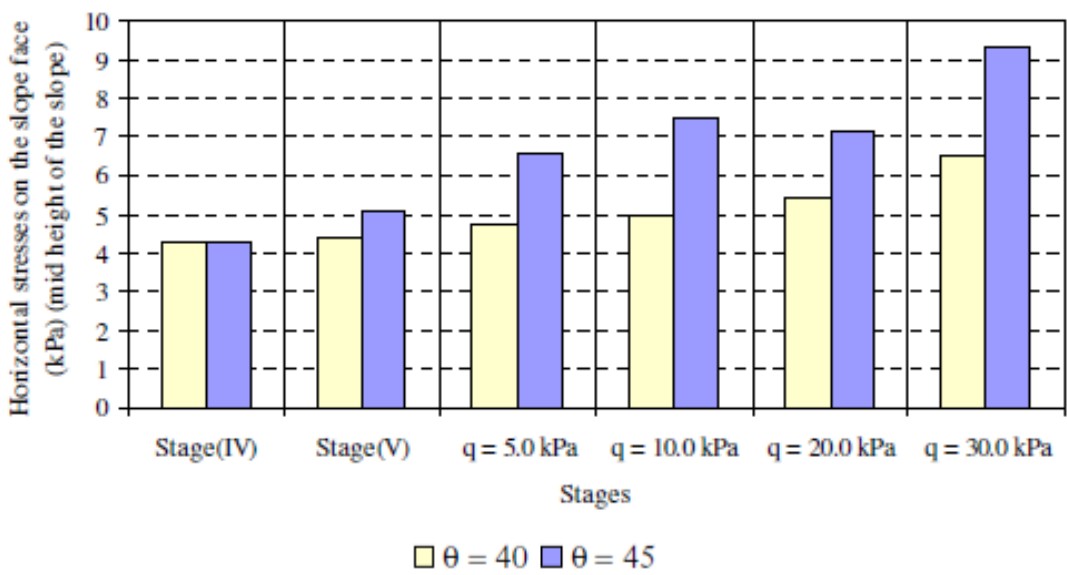

**Figure 15.** Influence of slope inclination on the slope face horizontal stresses at mid-height of slope.

### 3.2.5. Vertical Pressure under Nailed Soil Mass

The influence of the slope inclination on the vertical stresses under the nailed soil mass during construction stages and under surcharge loading is presented in Figure 16. It can be concluded from the figure that the effect of slope inclination on vertical stresses under the soil mass was directly proportional in loading stages, i.e., the higher the inclination angle, the higher the vertical stress under the soil mass. The vertical pressure during the slope construction will be the same at different angles of slope.

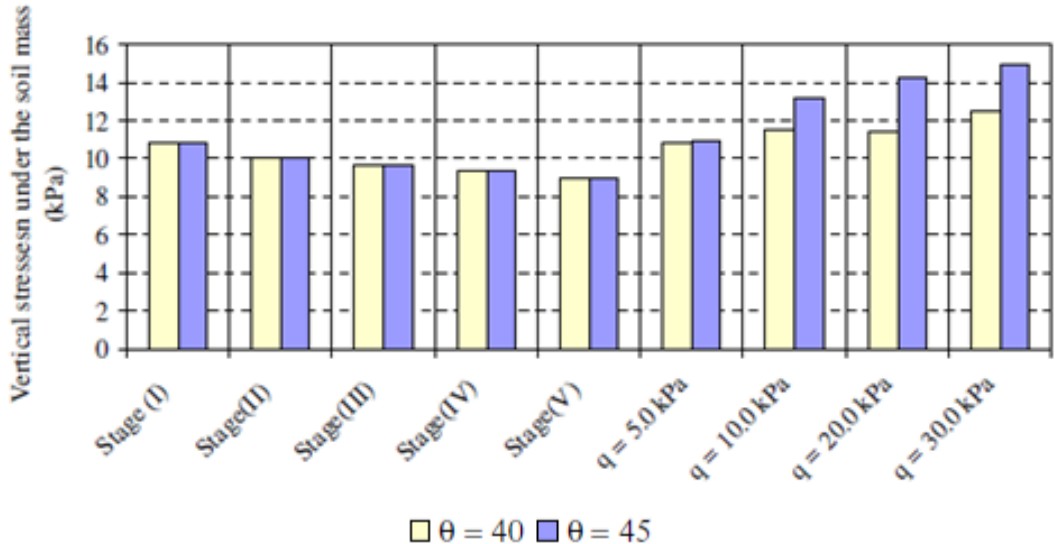

**Figure 16.** Influence of slope inclination on vertical stresses under the nailed soil mass.

### 3.2.6. Horizontal and Vertical Stresses behind Nailed Soil Mass

Figures 17 and 18 show the horizontal and vertical stresses behind the slopped soil mass in construction and loading stages measured at the middle of the slope. The figures depict that during excavation stages, the horizontal stresses behind the soil mass increase as the slope inclination decreases, whereas it decreases as the slope inclination decreases. The vertical stresses behind the soil mass decrease as the slope inclination decreases at all stages. The horizontal and vertical stresses behind the soil mass were increased when the surcharge load is increased.

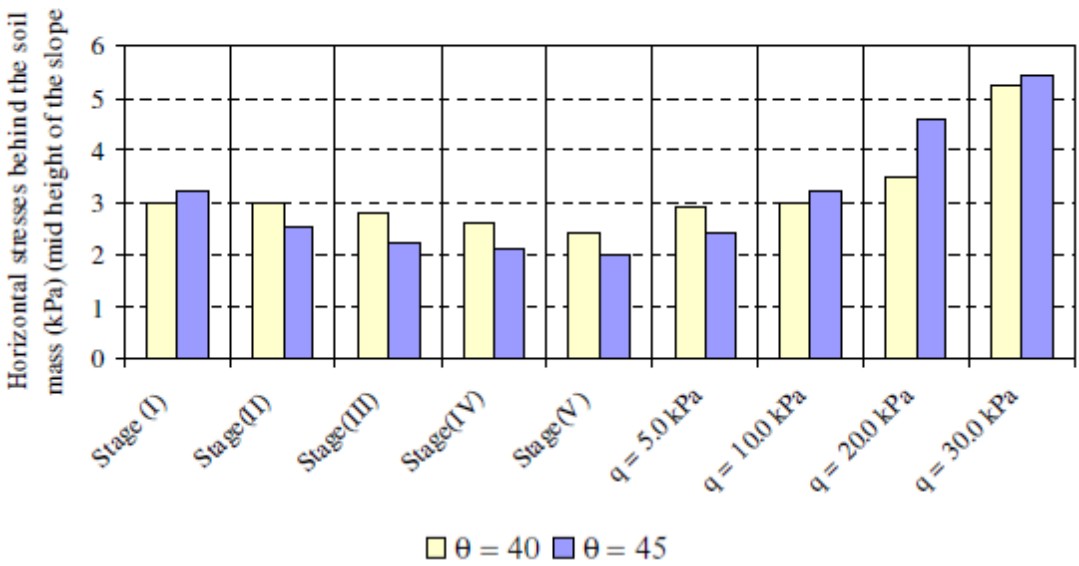

**Figure 17.** Influence of slope inclination on horizontal stresses behind the nailed soil mass at mid-height of slope.

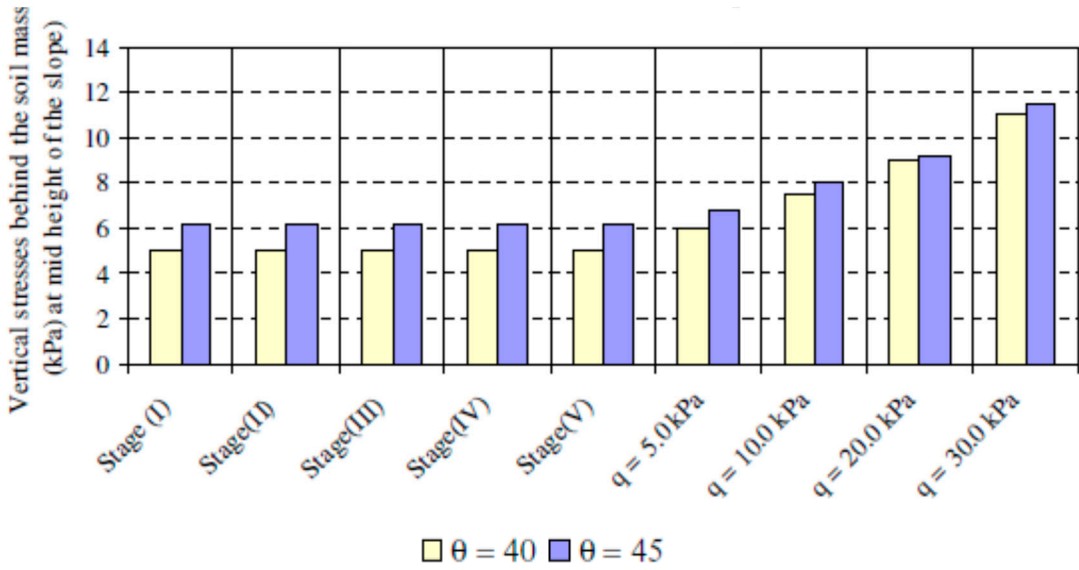

**Figure 18.** Influence of nail inclination on vertical stresses behind the nailed soil mass at mid-height of slope.

### 3.3. *Width of the Footing*

### 3.3.1. Lateral Displacement of the Soil-Nailed Slope

Figure 19 shows the effect of the footing width on the horizontal movement of the slope face during loading stages. The results indicate that at lower surcharge loading, there is no influence of footing width on the lateral movement of the soil-nailed slope, but with an increase of surcharge loading, lateral movement of the soil-nailed slope is increased. It

can also be concluded that the lateral movement of the facing increases with an increase of footing width.

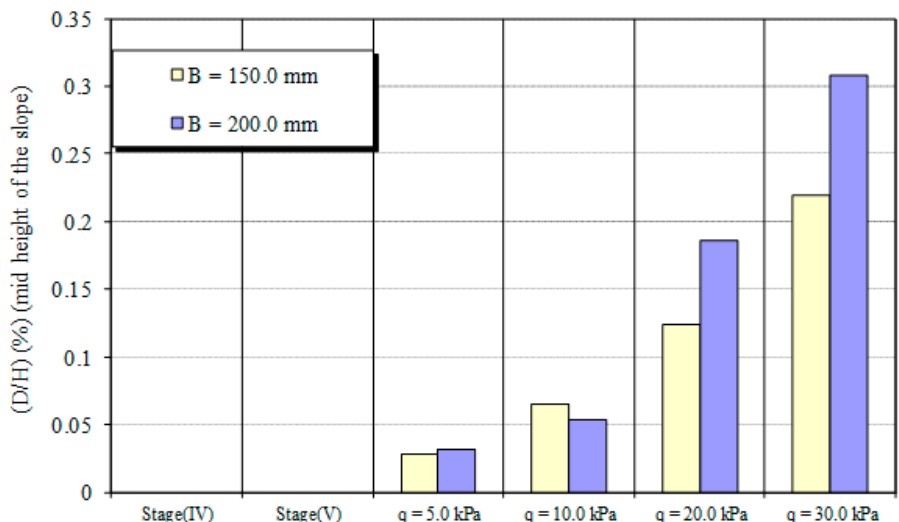

**Figure 19.** Influence of footing width on the horizontal movement at mid-height of slope.

### 3.3.2. Settlement of the Footing

The footing settlements for different footing widths at various footing pressures are given in Figure 20. It can be seen from the figure that the footing settlement increases when the footing width increases, and the percentage increase in footing settlement ranges from 20% to 38%. The increase of the footing settlement could be attributed to the increase of the lateral movement of the slope.

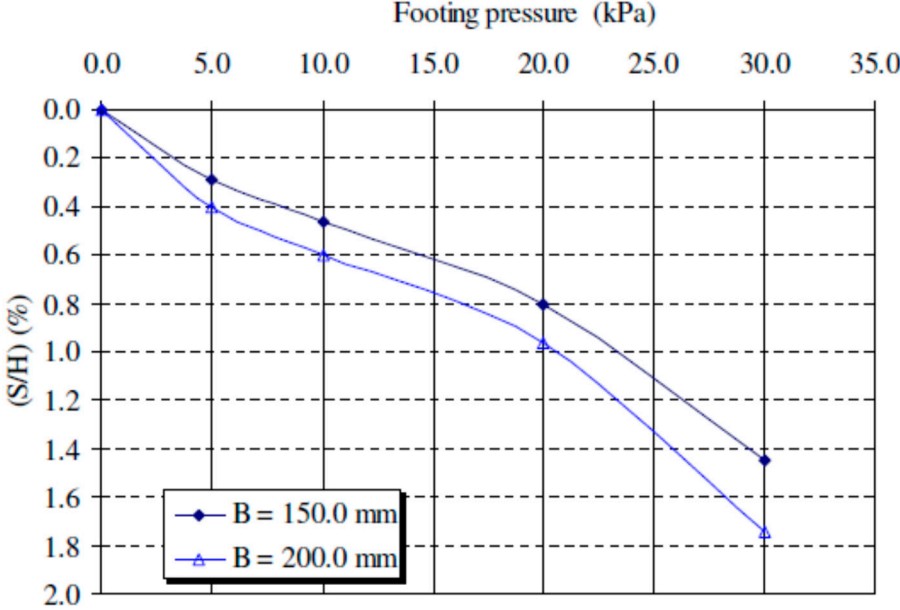

**Figure 20.** Influence of footing width on the settlement of the footing.

### 3.3.3. Force in the Nail

The influence of footing width on the maximum tensile force mobilized in the nails at different levels is presented in Figure 21. It can be concluded that the lower nails have lesser tensile force as compared to the upper nails at all stages of loading, and the tensile force increases in all nails as the footing pressure increases with different footing widths. The increase in footing width decreases the tensile force at lower surcharge (5 kPa), but with further increase of surcharge loadings, the tensile force in the nails is increased with

the increase of footing width because the stability of the slope at high footing width needs more tensile force mobilization.

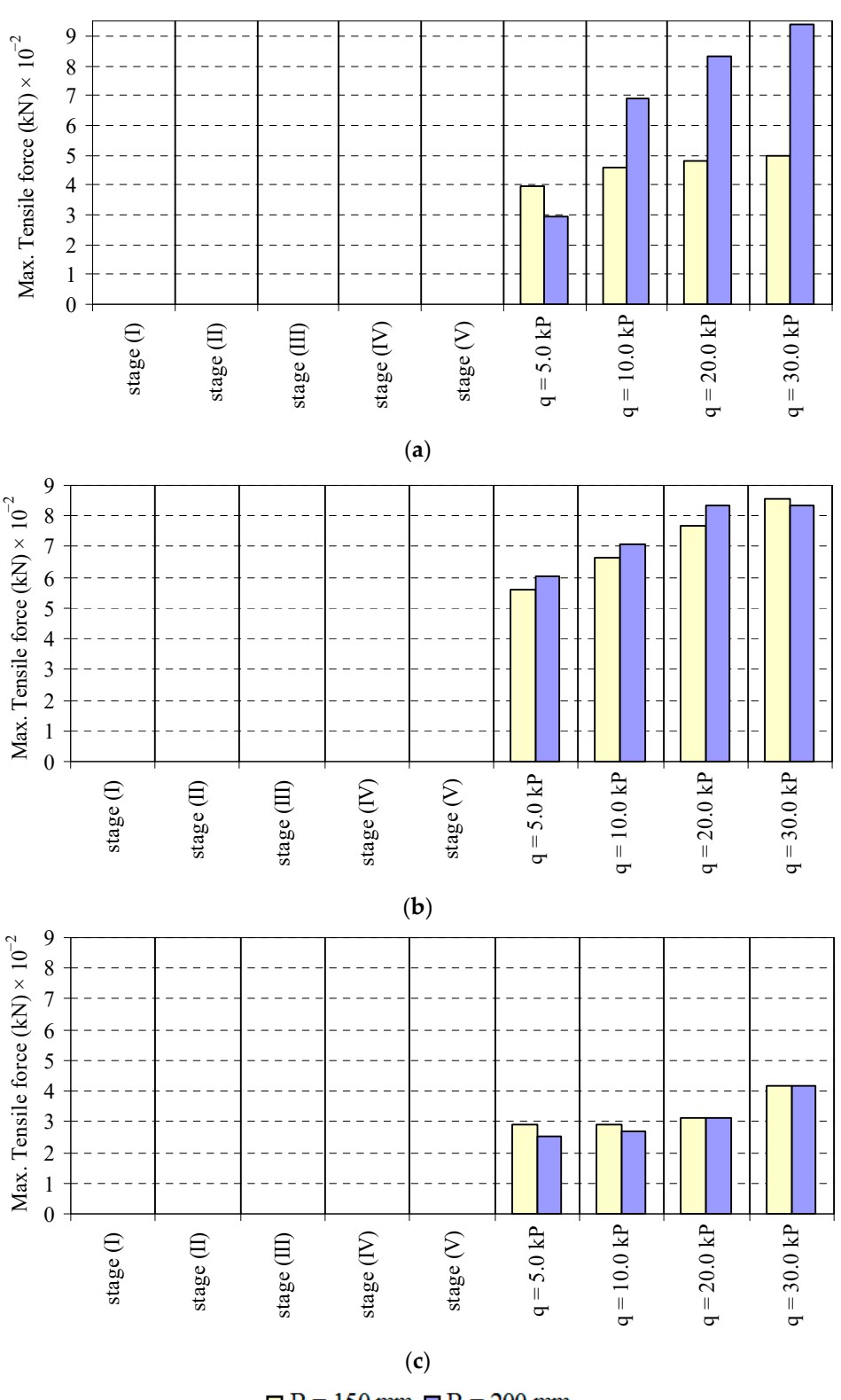

**Figure 21.** Influence of footing width on the maximum tensile force of the footing; (**a**) upper nail; (**b**) middle nail; (**c**) lower nail.

### 3.3.4. Horizontal Stresses on the Slope Face

Figure 22 gives the slope face horizontal stresses at mid-height of the slope with the different footing widths during the loading stages. The figure shows that the horizontal stresses at the slope face increase with the increase of the footing width and the footing pressure. The influence of the footing width is more at higher footing pressure. The percentage increase in the slope face horizontal stress with a 35% increase in footing width at the highest surcharge loading is considered to be about 20%.

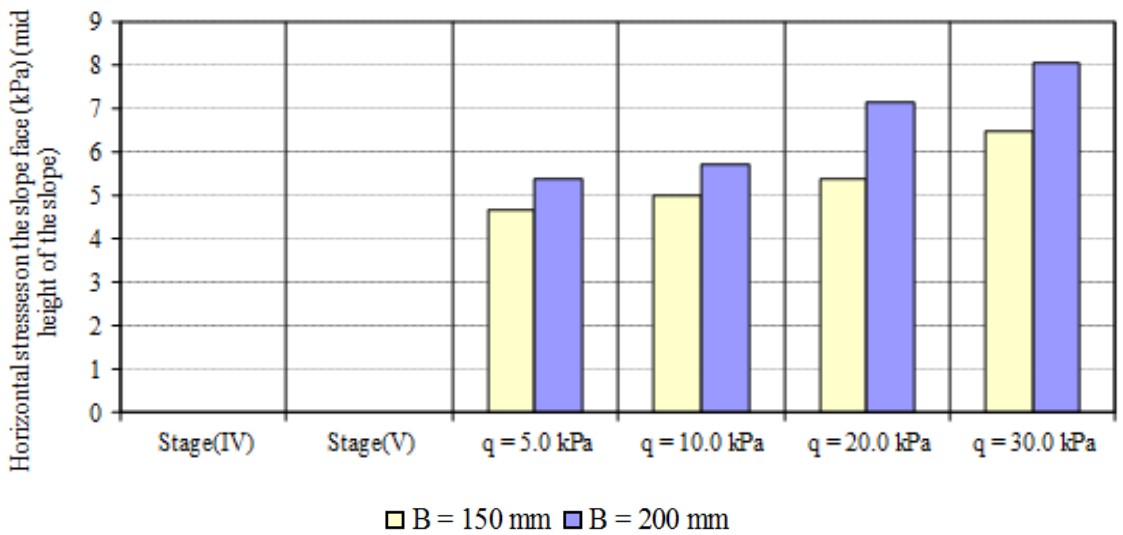

**Figure 22.** Influence of footing width on the slope face horizontal stresses at mid-height of slope.

### 3.3.5. Vertical Pressure under the Nailed Soil Mass

The vertical stresses under the nailed soil mass with the different footing widths under different stages are given in Figure 23. It can be seen from the figure that the effect of footing width on vertical stresses under the nailed soil mass was directly proportional, i.e., the higher the footing width, the higher the vertical stresses, and at different footing widths, increasing the surcharge load leads to increase in vertical stresses under the nailed soil mass.

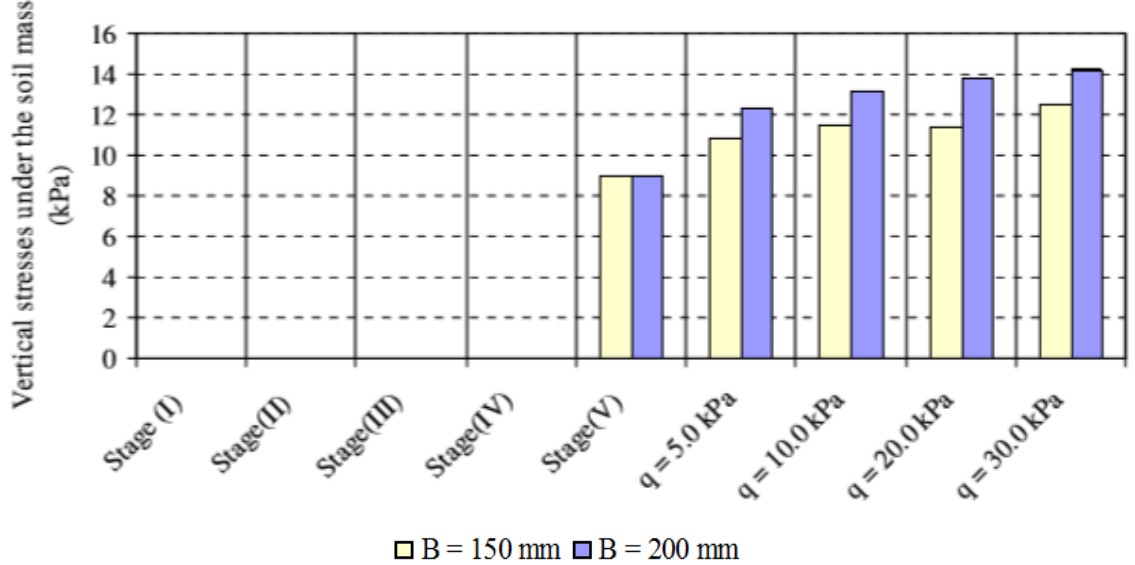

**Figure 23.** Influence of footing width on the slope face vertical stresses under the nails at mid-height of slope.

### 3.3.6. Horizontal and Vertical Stresses behind Nailed Soil Mass

Figures 24 and 25 present the effect of footing width on the horizontal and vertical stresses behind the soil mass in loading stages in the central portion of soil mass. The figures show that increasing footing width results in increasing the horizontal and vertical stresses behind the soil mass. The increase in the horizontal and vertical stresses behind the soil mass ranges from 0% to 21% and 0% to 57%, respectively, with a 35% increase in footing width and when surcharge load is increased from 5 kN/m² to 30 kN/m².

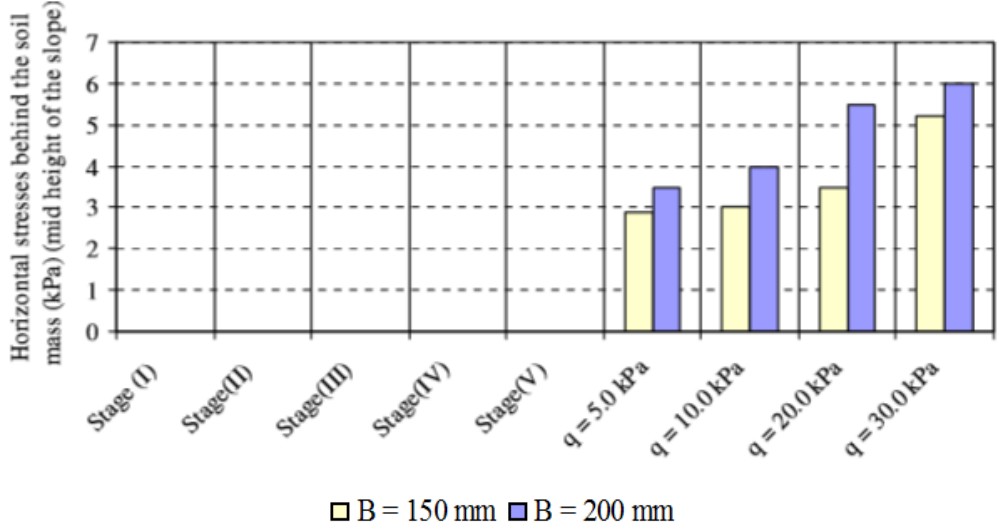

**Figure 24.** Influence of footing width on horizontal stresses behind the nailed soil mass at mid-height of slope.

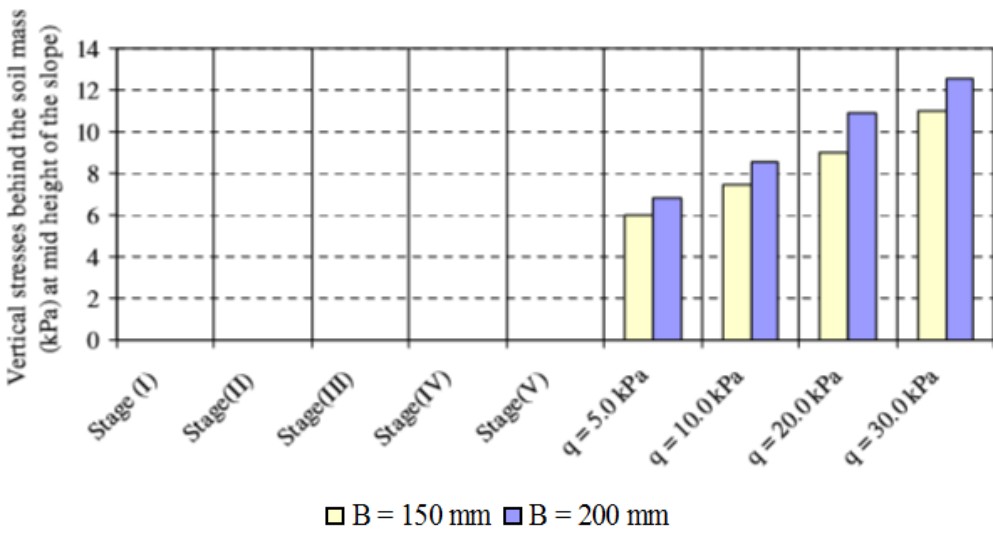

**Figure 25.** Influence of footing width on Vertical stresses behind the nailed soil mass at mid-height of slope.

### *3.4. Footing Position from Slope Face*

3.4.1. Horizontal Displacement of the Soil Nailing Slope

Figure 26 shows the effect of the footing location on the horizontal displacement of the slope face under the loading stage. It can be observed from the results that at the loading stage of footing pressure 20.0 kPa, with an increase of the footing distance from the slope face, the horizontal movement of the slope face at the mid-row of nails decreases. There is a sharp decrease in lateral movement of slope up to a footing distance of 1.5 B, and then a percentage decrease in slope movement is reduced. The reduction in slope-facing

displacement is about 82%, with an increase in the footing distance of twice the footing width.

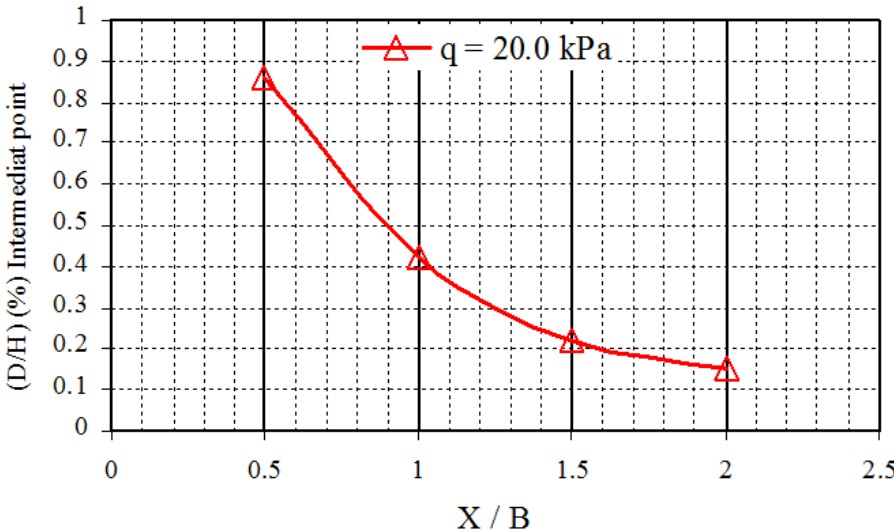

**Figure 26.** Influence of footing position on the mid-height slope horizontal movement.

3.4.2. Settlement of the Footing

Figure 27 presents the variation of the footing settlements for different footing locations under different footing pressures. From the figure, it can be seen that the footing settlement decreases as the footing distance from the slope face increases. A similar variation is obtained at all surcharge loads. This could be attributed to the fact that increasing the footing distance leads to a more stable slope.

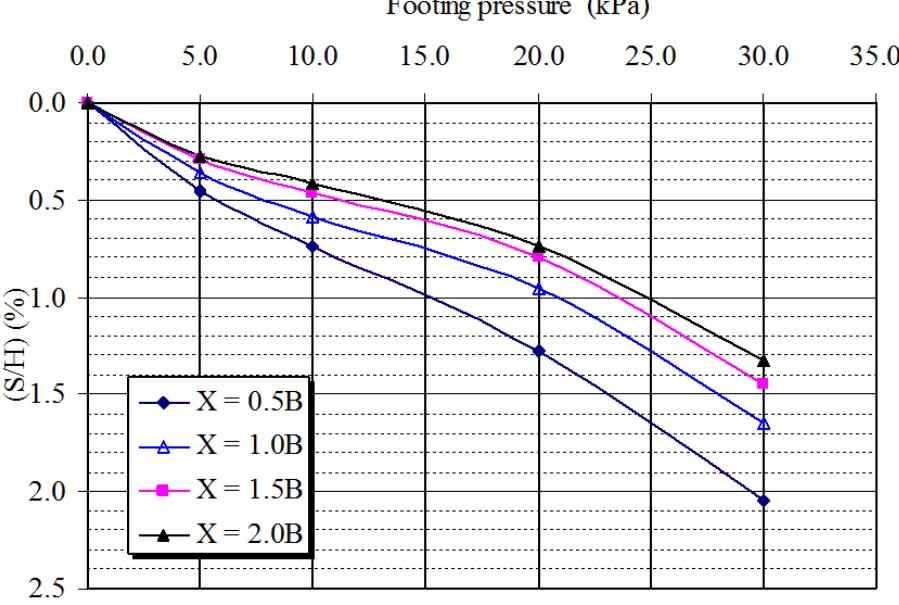

**Figure 27.** Influence of footing position on the settlement of the footing.

3.4.3. Force in the Nail

The variation of the footing position on the mobilized maximum tensile force in the nails at different levels is shown in Figure 28. The influence of footing position is small in the upper and lower nails. As the footing distance from the slope face increases, the maximum tensile force of the middle nails decreases. This may be attributed to the maximum horizontal displacements occurring at the mid-height of the slope.

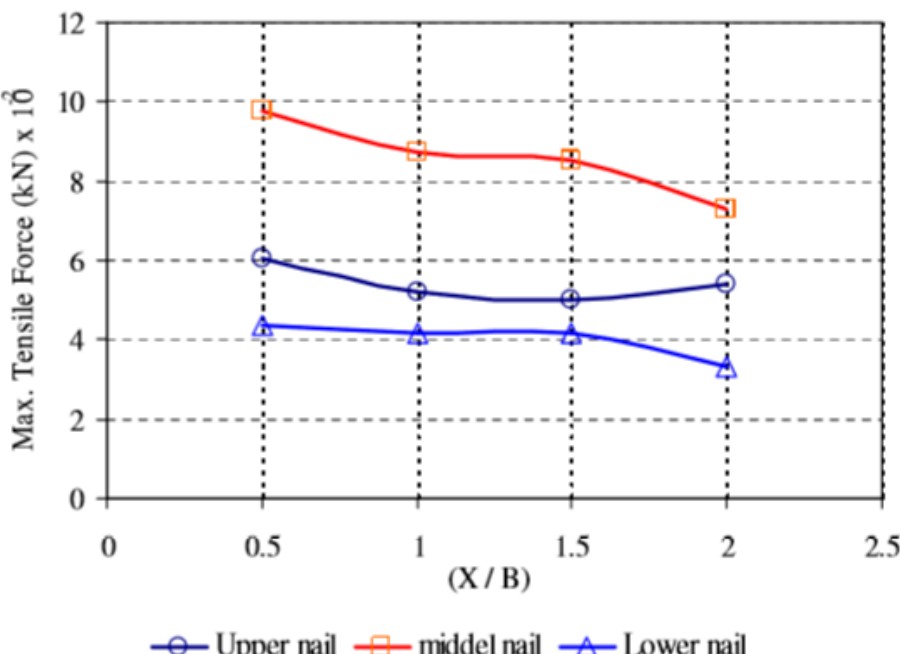

**Figure 28.** Influence of footing position on the maximum nail tensile force at levels.

### 3.4.4. Horizontal Stresses on the Slope Face

The variation of the footing position on the slope face horizontal stress at the middle of the slope during loading stages is shown in Figure 29. From the figure, it can be inferred that there are consider horizontal stresses present at the slope face, and the horizontal stresses measured at the same location of footing increase with the increases of footing pressure. As the footing distance from the slope face increases, the horizontal stresses at the slope face decrease as expected. By placing the footing to twice the footing width, the slope face horizontal stresses reduce to less than half of the original value.

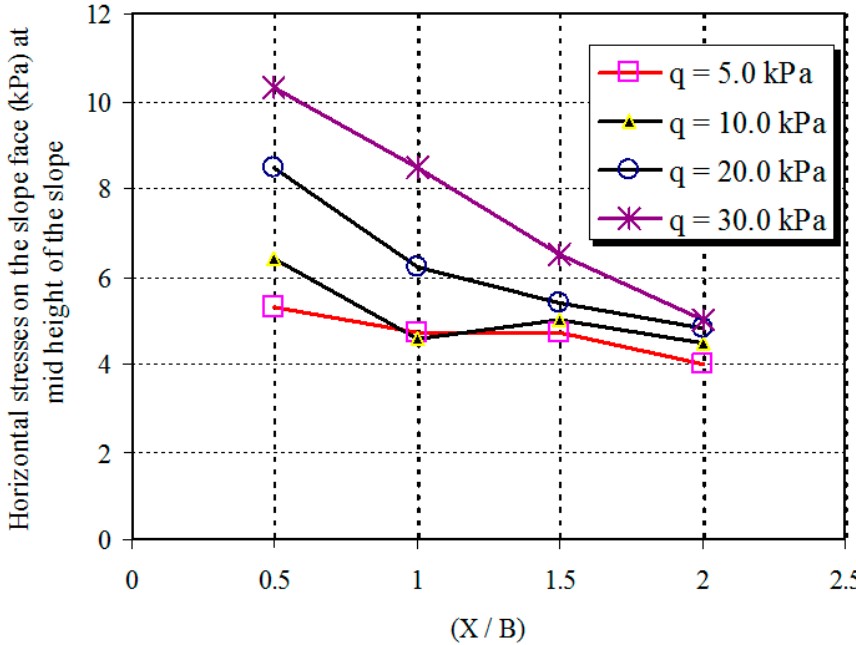

**Figure 29.** Influence of footing position on the slope face horizontal stresses under the nails at mid-height of slope.

### 3.4.5. Vertical Pressure under Nailed Soil Mass

Figure 30 presents the variation of the footing locations on the vertical stress under the soil mass with different footing pressure. From this figure, it can be seen that the effect of footing distance on vertical stresses under the soil mass is fluctuating, i.e., increasing the footing distance from slope face leads to an increase in vertical stresses under the soil mass up to a certain distance (x = 1.50 B) and a further increase of footing distance decreases the vertical stresses.

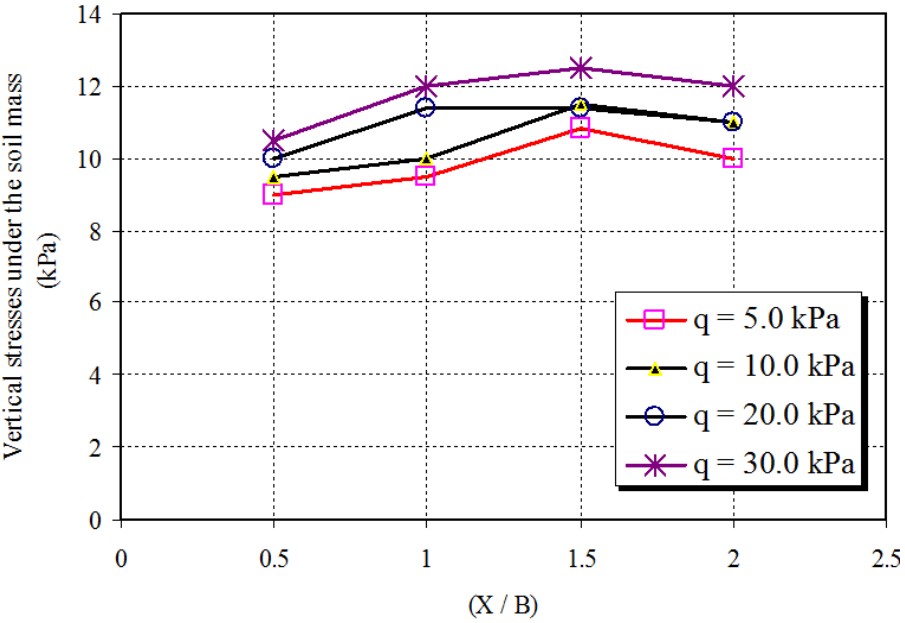

**Figure 30.** Influence of footing position on vertical stresses under the nailed soil mass.

### 3.4.6. Horizontal and Vertical Stresses behind Nailed Soil Mass

Figures 31 and 32 give the variation of the horizontal and vertical stresses behind the soil mass with different footing locations over the surface of the nailed slope with different foundation pressures. The influence on the horizontal and vertical stresses behind the soil mass with different footing locations is generally small at lower footing pressure. It is also clear from the figures that the horizontal stresses and vertical stresses behind the soil mass increase as the footing distance from the slope increases.

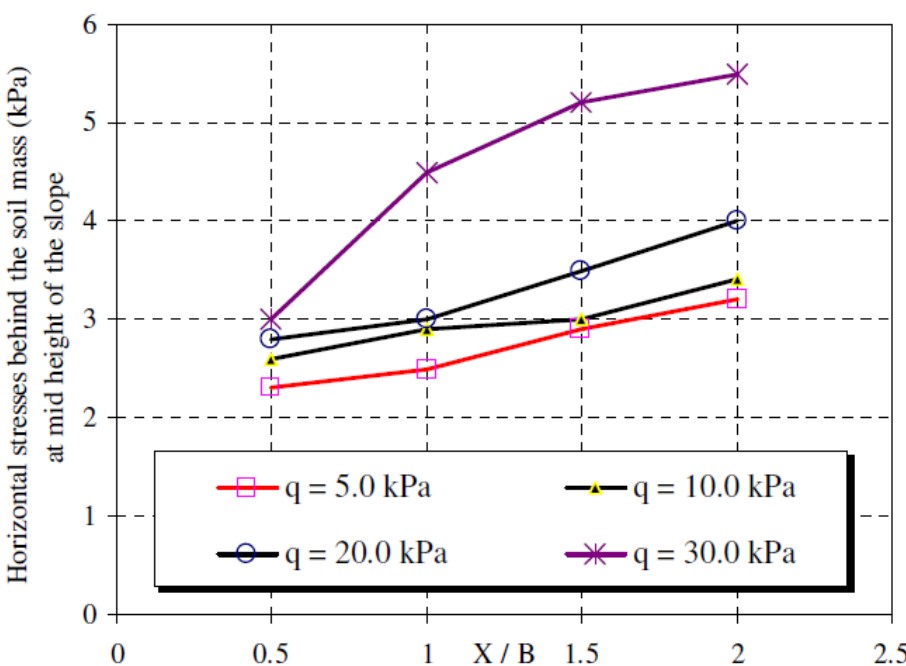

**Figure 31.** Influence of footing position on horizontal stresses behind the nailed soil mass at mid-height of slope.

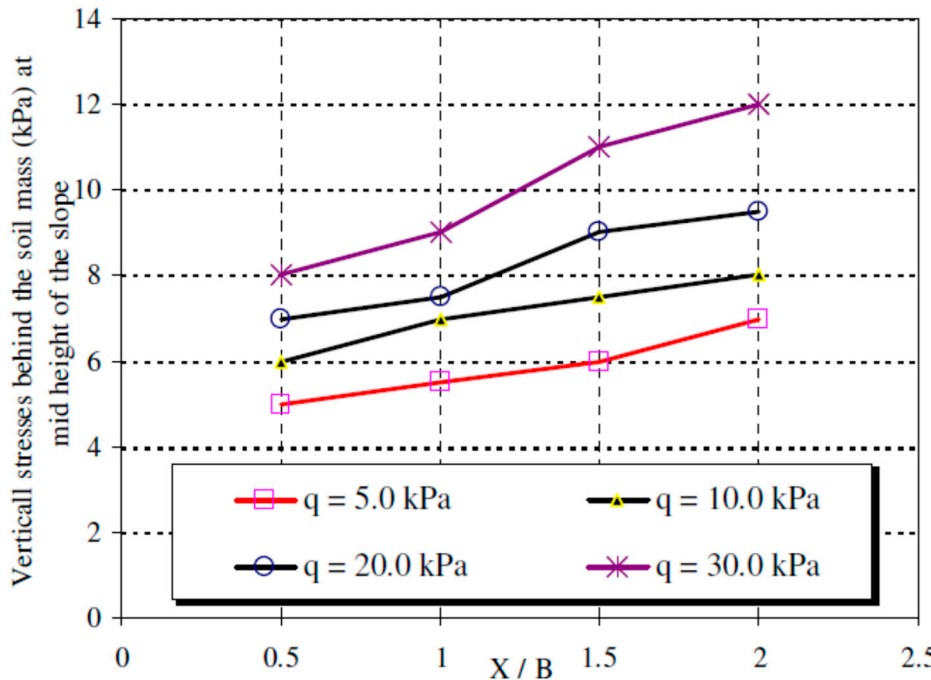

**Figure 32.** Influence of footing position on vertical stresses behind the nailed soil mass at mid-height of slope.

### 4. Numerical Modelling: Validation

The numerical model of earth slopes based on the finite element method is developed to ascertain the accuracy of laboratory model results. The PLAXIS-2D code is used to develop computational model of nailed soil slope considering plane strain conditions of soil slope. Four types of elements are used to model the sand, steel and Perspex glass material. A nonlinear elastic (Hyperbolic) model is used to simulate the sand material. In this model, the stress–strain curve is hyperbolic, and the soil modulus is a function of

confining stress and the shear stress that a soil is experiencing. Depending on the stress state and stress path, three soil moduli are required, namely, the initial modulus, the tangential modulus and the unloading–reloading modulus.

The geometry of the model is taken as 850 mm in height and 1760 mm in length, the inner dimensions of the sand bed used in the experimental model. The other nailed soil slope geometry parameters used in the model are taken from Figure 3. The geometry of the model is divided to meet the requirement of the construction stages and loading phases. The construction sequence is modeled in stages, as explained in Section 2.3.2 above. The 5 mm diameter steel bars with a length of 700 mm are modeled as the reinforcement, and a rigid steel plate of dimensions 840 mm × 150 mm × 22 mm is used to simulate the footing. Perspex plate with a thickness of 5.0 mm is selected to simulate the facing material, which is flexible enough to withstand the ground displacement during excavation. The strength parameters of nails, facing and footing materials are taken from Tables 2 and 3. The soil parameter used in the material model is given in Table 4. The discretized nailed soil slope model with boundary conditions is shown in Figure 33.

**Table 4.** Soil parameters of sand used in material model.

| Soil Type | Density (kN/m³) | Mohr-Coulomb Model | | | | Plastic Straining Due to Primary Comp./Deviator Loading $(E_{oed})^{ref}$ (kPa) | Elastic Unloading/Reloading | | Stress-Dependent Stiffness (m) |
|---|---|---|---|---|---|---|---|---|---|
| | | $\varphi$ | C (kPa) | $\psi$ | $\upsilon$ | | $(E_{ur})^{ref}$ (kPa) | $\upsilon_{ur}$ | |
| Loose sand (RD = 34%, e = 0.631) | 16.06 | 30 | 0.2 | 0 | 0.33 | 1327 | 3981 | 0.2 | 0.5 |
| Medium sand (RD = 48%, e = 0.597) | 16.41 | 34 | 0.2 | 4 | 0.31 | 1959 | 5877 | 0.2 | 0.5 |
| Dense sand (RD = 68%, e = 0.549) | 16.92 | 40 | 0.2 | 10 | 0.26 | 2632 | 7896 | 0.2 | 0.5 |

$\varphi$—friction angle; C—cohesion; $\psi$—dilatancy angle; $\upsilon$—Poisson's ratio; e—void ratio.

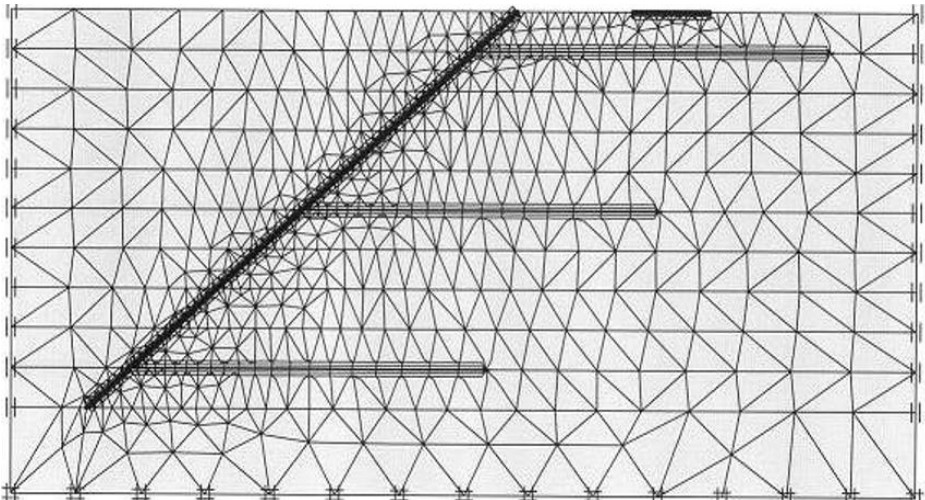

**Figure 33.** Triangular element discretization of nailed soil slope model.

To verify the accuracy of the laboratory results, a numerical model of nailed soil slope of height (*H*) 700 mm is developed with a length of horizontal nails as *H* and vertical spacing of nails as 0.4 *H*. The footing settlements (S/*H*, %) at different soil densities are computed using the developed numerical model. The plots of computational and experimental results of sand having three different relative densities (i.e., 35%—Fine sand, 48%—Medium sand and 68%—Dense sand) at different surcharge pressure, i.e., at the excavation stage and loading stage, are depicted in Figure 34. It is clear from the figure that the maximum footing

settlement occurred at a surcharge load exerting a foundation pressure of 30.0 kN/m$^2$ with both models. Moreover, the minimum footing settlement occurred at a surcharge load exerting a foundation pressure of 5.00 kN/m$^2$. It can be observed that the footing settlements obtained from the numerical model were overestimated compared to those obtained from the laboratory model for all cases of sand used. The percentage of difference in the settlement between experimental and numerical models ranged from 4.37% to 18.72% in the case of loose sand, whereas it ranged from 13.37% to 40.59% in the case of medium sand and it ranged from 0.39% to 35.02% in case of dense sand.

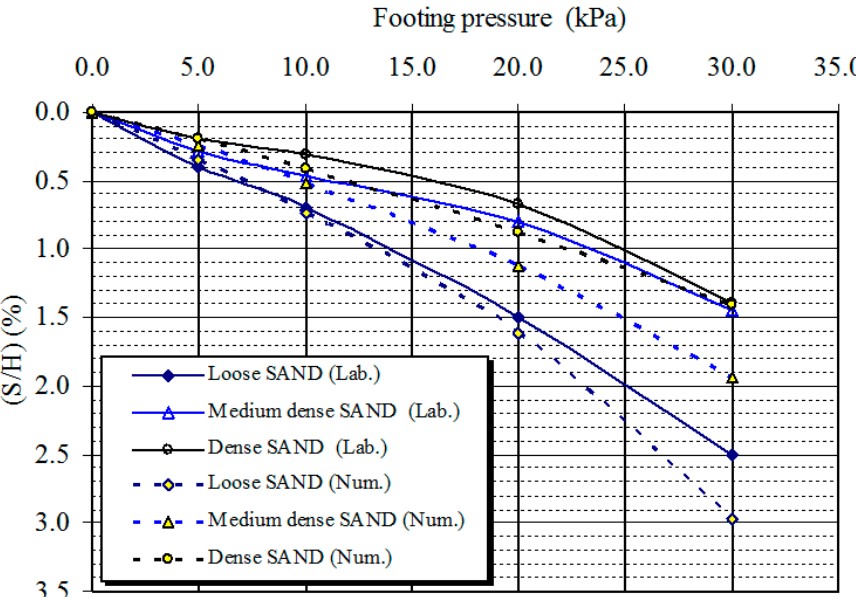

**Figure 34.** Footing settlement from laboratory and numerical models of nailed soil slope.

Figure 35 presents the vertical stresses under the nailed soil mass at excavation and loading stages obtained using the laboratory and numerical models at different soil densities. It is inferred from both model results that the denser the soil, the higher the vertical stresses under the soil mass. However, the vertical stresses are overestimated in numerical models as compared to laboratory models.

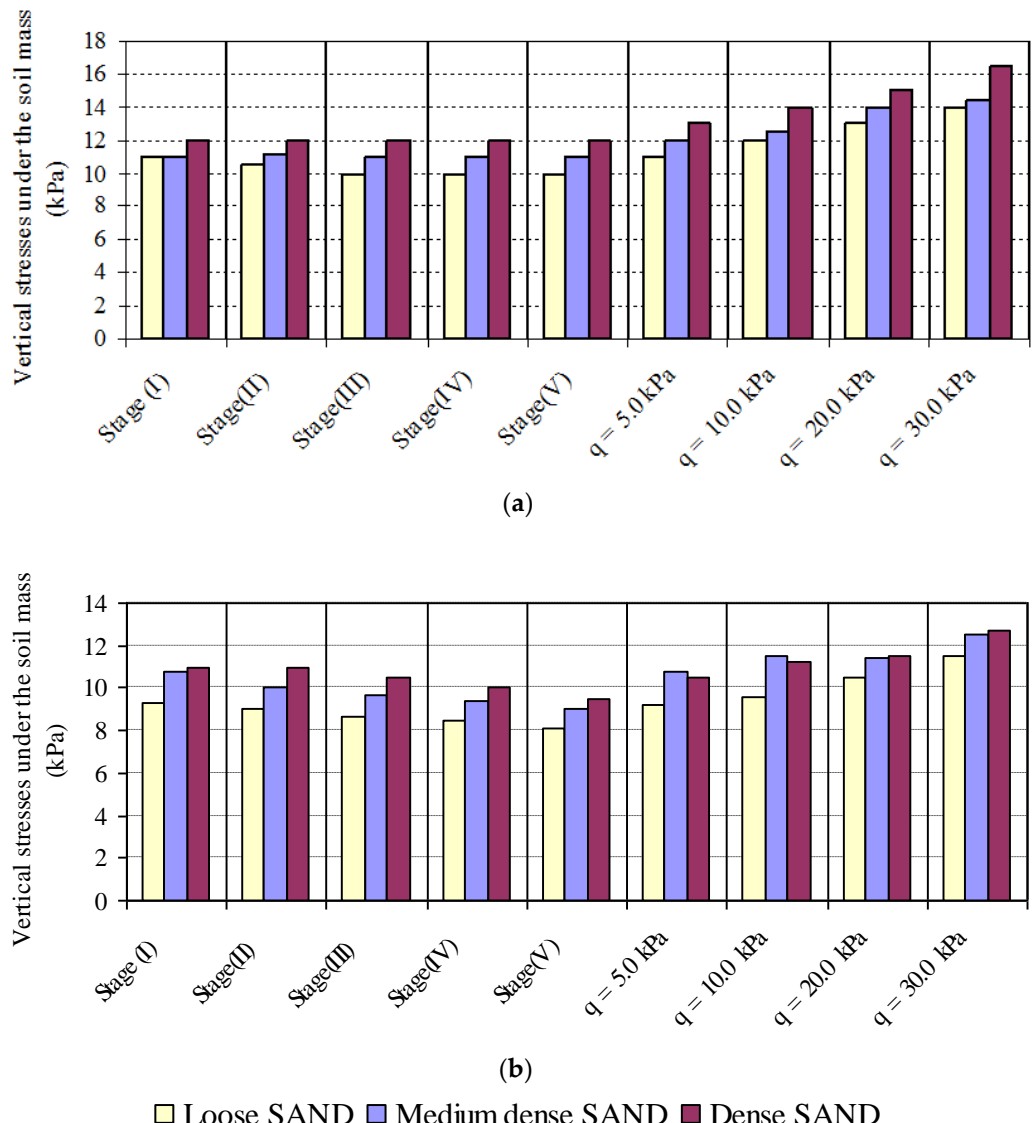

**Figure 35.** Effect of soil density on the vertical stresses under the nailed soil mass at mid-point; (**a**) numerical model; (**b**) laboratory model.

## 5. Conclusions

Soil nailing system is a ground improvement method used to stabilize the soil slopes. The study presents the laboratory investigation of earth nailing system models of non-cohesive soil having various soil and building foundation parameters. The laboratory models are prepared to have three soil types with different relative densities, i.e., 35%, 48%, 69%, two angles of slope, i.e., 40°, 45°, two footing widths, i.e., 150 mm and 200 mm and four positions of building foundation from the upper slope corner, i.e., 75 mm (0.5 B), 150 mm (B), 225 mm (1.5 B), 300 mm (2 B). The displacement of the slope, force in the nail, settlement of the footing and the earth pressure in back-fill soil due to different soil and footing parameters (relative density, slope angle, footing width, footing position) are measured.

The following conclusions may be drawn from the present study:

- The maximum lateral displacement and slope face horizontal pressures occur in the middle third and minimum lateral displacement, and slope face horizontal pressures occur in the lower third of slope with different soil and footing parameters under various footing pressures.

- The lateral movement of slope and footing settlements reduces with an increase of soil density and decreases with an inclination of soil slope. The increase in footing width increases the slope face displacement and footing settlement. The increase in strip footing distance from slope crest reduces the slope face displacement and footing settlement.
- The vertical and horizontal pressures behind the soil mass are unaffected during the construction process but begin to increase with the increase in the footing pressure. The distribution of vertical stress under the soil mass away from the slope part remains constant during various construction stages, and under different loading stages, the vertical stress under the soil mass is increased with an increase in foundation pressure.
- A decrease in soil density leads to an increase in maximum tensile force mobilized in the nails. Increasing the soil slope results in increasing the maximum tensile force of nails at the middle third of the soil slope, while soil slope increase results in a decrease in the maximum tensile force of nails at the lower and upper third of the soil slope. The mobilized tensile forces in nails are increased due to an increase in soil density and a decrease in footing distance from the slope face.
- The horizontal stresses at the slope face decrease with the decrease of relative density of soil while they are increased due to an increase in soil slope angle. The horizontal stresses at the slope face increase with the increase of footing width and decrease of footing distance from the slope crest.
- The vertical stress under the soil mass is more in high slope soil, denser soil, larger footing and greater distance from slop. The increase in soil density and slope angle results in an increase in vertical stress behind the soil mass and a decrease in the horizontal stress behind the soil mass with the increase of soil density and slope angle.
- Increasing the footing width leads to an increase in the vertical and horizontal stress behind the soil mass. The vertical and horizontal stresses behind the soil mass decrease with the decrease of footing distance from the slope crest.

**Author Contributions:** Conceptualization, M.H.M. and M.A.; methodology, M.H.M. and M.A.; resources, J.M.; writing—original draft preparation, M.H.M. and M.A.; writing—review and editing, J.M.; project administration, M.A. and J.M.; funding acquisition, M.A. All authors have read and agreed to the published version of the manuscript.

**Funding:** Funding for this research was given under award numbers R.G.P2/73/41 by the Deanship of Scientific Research; King Khalid University, Ministry of Education, Kingdom of Saudi Arabia.

**Institutional Review Board Statement:** Not Applicable.

**Informed Consent Statement:** Not Applicable.

**Acknowledgments:** The authors extend their appreciation to the Deanship of Scientific Research at King Khalid University for funding this work through a General Research Project under grant number (R.G.P2/73/41).

**Conflicts of Interest:** The authors declare no conflict of interest.

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
