# Peer review of "An Experimental Study of Nailed Soil Slope Models: Effects of Building Foundation and Soil Characteristics"

_applsci, doi:10.3390/app11167735_

Round 1
Reviewer 1 Report
The subject discussed is very interesting from a technical point of view, but unfortunately it does not contribute anything new either from a technical or scientific point of view, since what they show us are only the usual techniques when carrying out the excavation casting underground floors intended for basement floors.
The manuscript would present some type of technical interest if it would address how to deal with excavations with dividing buildings with the same number of floors below grade, or if it would address the presence of affected dividing services (as is the case of underground collectors of collection d epluviales, metro, etc,).
Regarding the structure of the manuscript, I consider it correct despite the fact that the conclusions obtained and the discussion of the manuscript are widely known topics.
Reviewer 2 Report
The manuscript is very interesting and it is very well written. However, there are a few things that need to be corrected:
- according to the international system, there should be a space between the number and the respective unit: eg.g. 2 L;
- why using capital letters in many words? e.g.: Sand, Layer; fine Gravel...etc.
- Figures 1 should be improved;
- Figure 3 have no quality.
- Figures of section 3.1.3 should have the number of the figure (Figure 6 a, b c, etc), legends have poor quality/definition. The same for Figures 7, 8, 9.
Reviewer 3 Report
The reviewer respect the authors efforts on doing these model tests of nailed slope. There are a lot of cases, and It might be very hard.
However, the target of the research is not clear. It is obvious that different level of deformation may be observed in different conditions. And the test results are basically reasonable. In other words, qualitatively obvious and nothing new. Thus, there is nothing interesting for readers.
To discuss the results quantitatively, important information is missing in the paper. For example, there is no scale on the soil tank shown in Fig.1. Fig.3 may be the cross section of the model, however, there are no explanation of 3D shape of the model. A nail is a bar, not a plate. So, it is impossible to make "plane strain condition (l.106)". However, 3d shape of the model cannot be understand by readers. Moreover, the shape of facing is also not clear. What is width 140 mm (less than footing width 150 mm) of facing shown in Table. 3. Fig.3 shall be a key to show the target, but the details are quite unclear.
There are many test data, shown by bars, or sometimes by plots. As the first showing of raw data, it is OK. However, there are no careful consideration and discussion on the results. Thus, again, the readers cannot understand the target of the research. For example, to discuss vertical pressure increase "due to increase in the unit weight (l.248)", why the authors did not calculate estimates by the depth and unit weight? The reviewer thinks the effect of compaction on vertical/horizontal pressure shall be carefully considered.
With these careful consideration, the discussion on why "middle nails have the maximum (l.303)" can be possible. But there are no discussion on this (a bit) interesting finding.
Not only for the test procedure, but also for the analysis, important information is missing. There are 3 levels of soil densities, but Table 4 shows only 1 case. In addition, the explanation of the parameters and constitutive models are missing. I think it is better to discuss more simple model first, i.e. rigid-plastic formulation to calculate earth pressure on the facing, or bearing capacity of the footings.
Reviewer 4 Report
This manuscript investigated the behavior of the reinforcing slope of soil nailing according to the change of the relative density of sand, the slope, and footing width through a series of model experiments. As a whole, this manuscript is well organized and written with the model experiment process and data. However, few comments need to be addressed as follows: Since the shear strength of sand changes depending on the relative density, please describe the internal friction angle in Tables 1 and 4 under which relative density conditions.
Round 2
Reviewer 3 Report
This is a research on nailed soil slope. The reviewer respect all the hard work of authors on the experiments. The explanation of test procedure was improved and the results are beneficial for readers, although those results are not surprisingly something new.
Although the physical property of soils are shown in table 1, there are no lab test results on soils. So, the information of shear strength, shear stiffness, etc. on soils in different relative densities are missing.
The construction process shown in Figure 2 is quite important. However, the meaning of black lines (2 lines in stage 1, and disappeared in stage 6) in soils are unclear.
In the test results, the fact that middle nails has maximum tensile force is interesting. Of course, as the author mentioned, it was come from the maximum deformation at the mid height of the slope. But the question is, why the mid height has the maximum displacement. It depends on the boundary condition, constraint condition, and construction process. Without the detailed consideration on the test results, the paper is just a technical report and not a research paper.
Analysis section was also improved. But the readers cannot understand why Perspex glass material was used. Also, how the excavation was modeled in the analysis was not mentioned clearly. What is worse, only the deformation was compared with the test results, and no explanation of ne tension forces on nails etc. were found. Thus, the reviewer cannot understand the purpose of this analysis section.
In conclusion, there are many summary of numbers on percentages. But these numbers depends on test conditions and similitude. Thus, the meanings of absolute values were not clear. Without detailed discussion on the reliability, it may be not appropriate showing numbers in conclusions.
Round 3
Reviewer 3 Report
With the discussion of the effect of the foundation as the boundary condition at the top of the slope, ,which are added now, I can understand the meaning of the research. However, the absence of the detailed lab test as the soil characteristics (various differences induced by the different level of relative density) cannot make the scientific soundness so high.